# Neural Methods for Point-wise Dependency Estimation

**Yao-Hung Hubert Tsai**[1], **Han Zhao**[2,*],
**Makoto Yamada**[34], **Louis-Philippe Morency**[1], **Ruslan Salakhutdinov**[1]
[1]Carnegie Mellon University, [2]D.E. Shaw & Co., [3] Kyoto University, [4]RIKEN AIP

## Abstract

Since its inception, the neural estimation of mutual information (MI) has demonstrated the empirical success of modeling expected dependency between high-dimensional random variables. However, MI is an aggregate statistic and cannot be used to measure point-wise dependency between different events. In this work, instead of estimating the expected dependency, we focus on estimating point-wise dependency (PD), which quantitatively measures how likely two outcomes co-occur. We show that we can naturally obtain PD when we are optimizing MI neural variational bounds. However, optimizing these bounds is challenging due to its large variance in practice. To address this issue, we develop two methods (free of optimizing MI variational bounds): Probabilistic Classifier and Density-Ratio Fitting.We demonstrate the effectiveness of our approaches in 1) MI estimation, 2) self-supervised representation learning, and 3) cross-modal retrieval task.

## 1 Introduction

Mutual Information (MI) measures the average statistical dependency between two random variables, and it has found abundant applications in practice, such as feature selection [12, 39], interpretable factor discovery [14, 46], genetic association studies [50], to name a few. Recent work [9, 40] proposed to use neural networks with gradient descent to estimate MI, which empirically scales better in high-dimension settings as compared to classic approaches (e.g., Kraskov (KSG) [28] estimator), which are known to suffer from the curse of dimensionality. Inspired by this line of work, we take a step further to present neural methods for point-wise dependency (PD) estimation. At a colloquial level, PD serves to understand the instance-level dependency between a pair of events taken by two random variables, which gives us a fine-grained understanding of the outcome. Formally, it can be realized as the ratio between likelihood of their co-occurrence to the likelihood of the product events: $p(x, y)/p(x)p(y)$ with $x$ and $y$ being the corresponding outcomes.

At first glance, it may seem straightforward to estimate PD by adopting prior density ratio estimation approaches [43, 44] to directly calculate the ratio between $p(x, y)$ and $p(x)p(y)$. Nonetheless, for the sake of tractability, previous methods are mainly kernel-based approaches that might be inadequate to scale to high-dimensional and complex-structured data. In this work, we introduce approaches for PD estimation that leverage the recent advances in rich and flexible neural networks. We show that we can naturally obtain PD when we are optimizing MI neural variational bounds [9, 40]. However, estimating these MI bounds often results in inevitably large variance [41]. To address this concern, we develop two data-driven approaches: *Probabilistic Classifier* and *Density-Ratio Fitting*. *Probabilistic Classifier* turns PD estimation into a supervised binary classification task, where we train a classifier to distinguish the observed joint distribution from the product of marginal distribution. This approach adopts cross-entropy loss using neural networks, which is favorable for optimization and exhibits a stable training trajectory with less variance. *Density-Ratio Fitting* seeks to minimize the least-square difference between the true and the estimated PD. Its objective involves no logarithm and exponentiation; hence, it is practically preferable due to its numerical stability.

We empirically analyze the advantages of PD neural estimation on three applications. First, we cast the challenging MI estimation problem to be a PD estimation problem. The re-formulation bypasses calculating MI lower bounds in prior work [9, 40], which suffers from large variance [41] in practice. Our empirical results demonstrate the low variance and bias of the proposed approach when comparing to prior MI neural estimators. Second, our PD estimation objectives also inspire new losses for contrastive self-supervised representation learning. Surprisingly, *Density-Ratio Fitting* inspired loss results in a consistent improvement over prior work in both shallow [48] and deep [5] neural architectures. Third, we study the use of PD estimation for data containing information across modalities. More specifically, we analyze the cross-modal retrieval task on human speech and text corpora. We make our experiments publicly available at `https://github.com/yaohungt/Pointwise_Dependency_Neural_Estimation`.

## 2  Related Work

**Point-wise Dependency Estimation**  Prior literature studies point-wise dependency (PD) with three groups of estimation methods: *counting-based* [10, 16, 31], *kernel-based* [49], and *likelihood-based* [32]. *Counting-based* methods approximate the joint density by counting the occurrence of the pair (i.e., $(x, y)$) and the marginal density by counting the presence of the individual outcome (i.e., $x$ or $y$). Counting based approaches can only work on discrete data and may be unrealistic when the data is sparse. *Kernel-based* method, particularly pointwise HSIC [49], can be seen as a smoothed variant of the counting-based methods, which adopts the kernel to measure the similarity between sparse data. Although this method manifests nice robustness to sparse data, its computational cost is high with high-dimensional data. *Likelihood-based* approaches instead approximate conditional likelihood (i.e., $p(y|x)$) and marginal likelihood (i.e., $p(y)$) using function approximators such as neural networks. Although this approach can be adapted to continuous data, it involves marginal likelihood estimation, which is challenging [18, 25] and may perform poorly in practice. On the other hand, our presented approaches involve no marginal likelihood estimation, can work on both discrete and continuous data, and leverage neural networks with gradient descent in high-dimensional settings.

**Density Ratio Estimation**  To calculate the ratio between densities ($p(x)/q(x)$), prior density ratio estimation approaches [43, 44] propose to estimate the ratio directly and avoid estimating the density ($p(x)$ and $q(x)$). For example, Sugiyama *et al.* [44] fit the true density ratio model under the Bregman divergence [11] and further develop a robust density estimation method under the power divergence [8]. While it is straightforward to apply these approaches to PD estimation, these approaches are studied in the context of kernel-based methods, which can make it difficult to apply in practice when data is high-dimensional and complex-structured. Our approaches contrarily take advantage of high-capacity neural networks.

**Neural Methods for Mutual Information Estimation**  Recent approaches [9, 40] present neural methods that estimate mutual information (MI) via its variational bounds. They consider MI 1) lower bounds such as Donsker-Varadhan bound [17] and Nguyen-Wainwright-Jordan bound [34]; and 2) upper bound such as Barber-Agakov bound [6]. These bounds exhibit inevitable large variance [41] and have severe training instability in practice [21, 48]. In our discussion, we show that we can obtain PD when optimizing these bounds. Additionally, we present alternative PD estimation methods that do not involve calculating MI variational bounds and are favorable in practice.

## 3  Point-wise Dependency Neural Estimation

Our paper aims to identify the association for a pair of outcomes $(x, y) \in \mathcal{X} \times \mathcal{Y}$ by studying their point-wise dependency. We use an uppercase letter to denote a random variable (i.e., $X$), a lowercase letter to indicate an outcome $x$ drawn from a particular distribution (i.e., $x \sim P_X$), and a calligraphy letter $\mathcal{X}$ to represent a sample space (i.e., $x \in \mathcal{X}$). The joint distribution of $X, Y$ is represented by $P_{X,Y}$, and the product of their marginals is represented by $P_X P_Y$. Throughout the paper, we use the conventional notation $I(X; Y)$ to denote the mutual information between random variables $X$ and $Y$.

Formally, we define the following point-wise dependency (PD) to quantitatively measure the discrepancy between *the probability of their co-occurrence* and *the probability of independent occurrences*.

**Definition 1** (Point-wise Dependency). Given a pair of outcomes $(x, y) \sim P_{X,Y}$, their point-wise dependency is defined as $r(x, y) := p(x, y)/p(x)p(y)$.

PD is non-negative. Intuitively, when $r(x, y) > 1$, it means $(x, y)$ co-occur more often than their independent occurances. Similarly, when $r(x, y) \leq 1$, it means they co-occur less frequently. Our goal is to estimate $r(x, y)$ by approximating it using neural network $\hat{r}_\theta(x, y)$ with parameter $\theta \in \Theta$.

### 3.1 Mutual Information and Point-wise Dependency

A related quantitative measurement of point-wise dependency is Point-wise mutual information (PMI) [10], which is the logarithm of PD (PMI $:= f(x, y) = \log r(x, y)$). In this subsection, we shall discuss parametrized estimation of PMI using neural networks $\hat{f}_\theta(x, y)$ with parameter $\theta$. By definition, mutual information $I(X; Y)$ is the expected value of PMI: $I(X; Y) = \mathbb{E}_P[\log r(X, Y)] = \mathbb{E}_P[f(X, Y)]$. Hence by using $\hat{f}_\theta$ as a plug-in, we can obtain an approximation of the mutual information with $\mathbb{E}_P[\hat{f}_\theta(X, Y)]$. Reversely, we will show that PMI can be obtained when optimizing MI (neural) variational bounds and present two methods to do so, one as unconstrained optimization and the other as constrained optimization problem.

**(Unconstrained Optimization) Variational Bounds of Mutual Information**  Recent work [9, 40] proposes to estimate MI using neural networks by exploiting either the variational MI lower bounds [9] or the variational MI form [40]. In particular, Belghazi *et al.* [9] proposed the $I_{\mathrm{DV}}$ estimator, standing for Donsker-Varadhan (DV) lower bound [17] of MI. On the other hand, Poole *et al.* [40] proposed the $I_{\mathrm{JS}}$ estimator, corresponding to using f-GAN objective [35] as a lower bound of Jensen-Shannon (JS) divergence between $P_{X,Y}$ and $P_X P_Y$. $I_{\mathrm{JS}}$ is found to be more stable then $I_{\mathrm{DV}}$ and other variational lower bounds, and thus it is widely used in prior work [21, 40, 41], defined as follows:

$$I_{\mathrm{JS}} := \sup_{\theta \in \Theta} \mathbb{E}_{P_{X,Y}}\Big[ -\operatorname{softplus}\Big( -\hat{f}_\theta(x, y)\Big)\Big] - \mathbb{E}_{P_X P_Y}\Big[\operatorname{softplus}\Big(\hat{f}_\theta(x, y)\Big)\Big], \qquad (1)$$

where we use $\operatorname{softplus}$ to denote $\operatorname{softplus}(x) = \log(1 + \exp(x))$. It could be readily verified that the optimal $\hat{f}_\theta^*(x, y) = \log(p(x, y)/p(x)p(y))$ [40]. We refer this objective as *Variational Bounds of Mutual Information* approach for PMI estimation.

**(Constrained Optimization) Density Matching**  This method considers to match the true joint density $p(x, y)$ and the estimated joint density $\hat{p}_\theta(x, y) := e^{\hat{f}_\theta(x,y)}p(x)p(y)$ by minimizing the following KL divergence:

$$\inf_{\theta \in \Theta} D_{\mathrm{KL}}(P_{X,Y} \| \hat{P}_{\theta X,Y}) := \inf_{\theta \in \Theta} I(X; Y) - \mathbb{E}_{P_{X,Y}}\Big[\hat{f}_\theta(x, y)\Big] \Leftrightarrow \sup_{\theta \in \Theta} \mathbb{E}_{P_{X,Y}}\Big[\hat{f}_\theta(x, y)\Big].$$

Since KL divergence has a minimum value of 0, it is easy to see that $\forall \theta \in \Theta$, $\mathbb{E}_{P_{X,Y}}[\hat{f}_\theta(x, y)]$ is a lower bound of MI. Note that this objective is a constrained optimization problem, since we need to ensure the estimated joint density is a valid density function: $\hat{p}_\theta(x, y) \geq 0$ and $\iint \hat{p}_\theta(x, y)\,\mathrm{d}x\mathrm{d}y = 1$. Equivalently, the constraints could be formed as $e^{\hat{f}_\theta(x,y)} \geq 0$ (trivially true) and $\mathbb{E}_{P_X P_Y}[e^{\hat{f}_\theta(x,y)}] = 1$. Putting everything together, we can reformulate the following constrained optimization problem:

$$\max_{\theta \in \Theta} \mathbb{E}_{P_{X,Y}}[\hat{f}_\theta(x, y)], \quad \text{subject to } \mathbb{E}_{P_X P_Y}[e^{\hat{f}_\theta(x,y)}] = 1,$$

which is also called KL importance estimation procedure [42] with a unique solution $\hat{f}_\theta^*(x, y) = \log(p(x, y)/p(x)p(y))$. The Lagrangian of the above constrained problem is

$$\max_{\theta \in \Theta} \mathbb{E}_{P_{X,Y}}[\hat{f}_\theta(x, y)] - \lambda \cdot \Big(\mathbb{E}_{P_X P_Y}[e^{\hat{f}_\theta(x,y)}] - 1\Big), \qquad (2)$$

where $\lambda \in \mathbb{R}$ is the dual variable. Furthermore, penalty method could also be used to transform the original constrained optimization problem to an unconstrained one:

$$\max_{\theta \in \Theta} \mathbb{E}_{P_{X,Y}}[\hat{f}_\theta(x, y)] - \eta \cdot \Big(\log \mathbb{E}_{P_X P_Y}[e^{\hat{f}_\theta(x,y)}]\Big)^2, \qquad (3)$$

where $\eta > 0$ is the penalty coefficient. We refer Eq. (2) as *Density Matching I* and Eq. (3) as *Density Matching II* for PMI estimation.

## 3.2 Proposed Methods for Point-wise Dependency (PD) Estimation

In the last section, we introduce how to obtain PMI by optimizing various MI variational bounds. In this section, instead of estimating PMI, we present two methods to estimate PD $(p(x,y)/p(x)p(y))$, i.e., the *Probabilistic Classifier* method and the *Density-Ratio Fitting* method. We argue that the presented PD estimation methods admit better training stability than the PMI estimation methods discussed in the last section. On the one hand, the Probabilistic Classifier method casts PD estimation as a binary classification task, where the binary cross-entropy loss can be used and optimized in existing optimization packages [1, 38]. On the other hand, the Density-Ratio Fitting method contains no logarithm or exponentiation, which are often the roots of the instability in MI (or PMI) estimation [40, 41]. In what follows, we present both methods in a sequel.

**Probabilistic Classifier Method**  This approach casts the PD estimation as the problem of estimating the 'class'-posterior probability. First, we use a Bernoulli random variable $C$ to classify the samples drawn from the joint density ($C = 1$ for $(x, y) \sim P_{X,Y}$) and the samples drawn from product of the marginal densities ($C = 0$ for $(x, y) \sim P_X P_Y$). Equivalently, the likelihood function $p(x, y \mid C = 1) := p(x, y)$ and $p(x, y \mid C = 0) := p(x)p(y)$. By Bayes' Theorem, we re-express PD by the ratio of two class-posterior probability:

$$r(x,y) = \frac{p(x,y)}{p(x)p(y)} = \frac{p(x, y \mid C = 1)}{p(x, y \mid C = 0)} = \frac{p(C = 0)}{p(C = 1)} \frac{p(C = 1 \mid x, y)}{p(C = 0 \mid x, y)}.$$

In the above equation, the ratio $\frac{p(C=0)}{p(C=1)}$ can be approximated by the ratio of the sample size:

$$\frac{\hat{p}(C = 0)}{\hat{p}(C = 1)} = \frac{(n_{P_X P_Y})/(n_{P_X P_Y} + n_{P_{X,Y}})}{(n_{P_{X,Y}})/(n_{P_X P_Y} + n_{P_{X,Y}})} = \frac{n_{P_X P_Y}}{n_{P_{X,Y}}},$$

and we use a probability classifier $\hat{p}_\theta(C \mid x, y)$ parameterized by a neural network $\theta$ to approximate the class-posterior classifier $p(C \mid x, y)$. By adopting the binary cross-entropy loss, the objective has the following form:

$$\max_{\theta \in \Theta} \mathbb{E}_{P_{X,Y}}[\log \hat{p}_\theta(C = 1 \mid x, y)] + \mathbb{E}_{P_X P_Y}[\log (1 - \hat{p}_\theta(C = 1 \mid x, y))]. \tag{4}$$

Then, bringing all the equations together, we obtain the *Probabilistic Classifier* PD estimator:

$$\hat{r}_\theta(x,y) = \frac{n_{P_X P_Y}}{n_{P_{X,Y}}} \frac{\hat{p}_\theta(C = 1 \mid x, y)}{\hat{p}_\theta(C = 0 \mid x, y)}, \quad \text{with } (x, y) \sim P_{X,Y} \text{ or } (x, y) \sim P_X P_Y. \tag{5}$$

**Density-Ratio Fitting Method**  This approach considers to minimize the expected least-square difference between the true PD $r(x, y)$ and the estimated PD $\hat{r}_\theta(x, y)$:

$$\inf_{\theta \in \Theta} \mathbb{E}_{P_X P_Y}[(r(x,y) - \hat{r}_\theta(x,y))^2] \Leftrightarrow \sup_{\theta \in \Theta} \mathbb{E}_{P_{X,Y}}[\hat{r}_\theta(x,y)] - \frac{1}{2}\mathbb{E}_{P_X P_Y}[\hat{r}_\theta^2(x,y)]. \tag{6}$$

The objective is also called least-square density-ratio fitting method [24] and has a unique solution $\hat{r}_\theta^*(x, y) = p(x, y)/p(x)p(y)$. We refer Eq. (6) as *Density-Ratio Fitting* PD estimation.

## 4 Application I: Mutual Information Estimation

By definition, as the average effect of point-wise dependency (PD), Mutual Information (MI) measures the statistical independence between random variables:

$$I(X;Y) = D_{\mathrm{KL}}(P_{X,Y} \parallel P_X P_Y) = \iint p(x,y) \log \frac{p(x,y)}{p(x)p(y)} \, \mathrm{d}x \mathrm{d}y = \mathbb{E}_{P_{X,Y}}[\log r(x,y)]$$
$$\approx \mathbb{E}_{P_{X,Y}}[\log \hat{r}_\theta(x,y)] \approx \mathbb{E}_{P_{X,Y}}[\hat{f}_\theta(x,y)], \tag{7}$$

where we estimate MI by directly plugging-in PD (i.e., $\hat{r}_\theta$ in Eq. (5), (6)) or PMI (i.e., $\hat{f}_\theta$ in Eq. (1), (2), and (3)). In summary, we cast the MI estimation problem to a PD or PMI estimation problem.

Table 1: MI neural estimation methods. The estimation procedure is dissected into learning and inference phases, which may use different objectives. Baselines consider to estimate MI via lower bounds, while ours consider to estimate MI via plugging in PD ($\hat{r}_\theta$) or PMI ($\hat{f}_\theta$) estimators.

| Baselines | Learning | Inference | Ours | Learning | Inference |
|---|---|---|---|---|---|
| CPC [36] | $I_{\text{CPC}}$ [36] | $I_{\text{CPC}}$ [36] | Variational MI Bounds | $I_{\text{JS}}$ [35] (Eq. (1)) | Eq. (7) with $\hat{f}_\theta$ |
| NWJ [9] | $I_{\text{NWJ}}$ [9, 34] | $I_{\text{NWJ}}$ [9, 34] | Probabilistic Classifier | Eq. (4) | Eq. (7) with $\hat{r}_\theta$ in Eq. (5) |
| JS [40] | $I_{\text{JS}}$ [35] (Eq. (1)) | $I_{\text{NWJ}}$ [9, 34] | Density Matching I | Eq. (2) | Eq. (7) with $\hat{f}_\theta$ |
| DV (MINE) [9] | $I_{\text{DV}}$ [9] | $I_{\text{DV}}$ [9, 17] | Density Matching II | Eq. (3) | Eq. (7) with $\hat{f}_\theta$ |
| SMILE [41] | $I_{\text{JS}}$ [35] (Eq. (1)) | $I_{\text{DV}}$ [9, 17] | Density-Ratio Fitting | Eq. (6) | Eq. (7) with $\hat{r}_\theta$ |

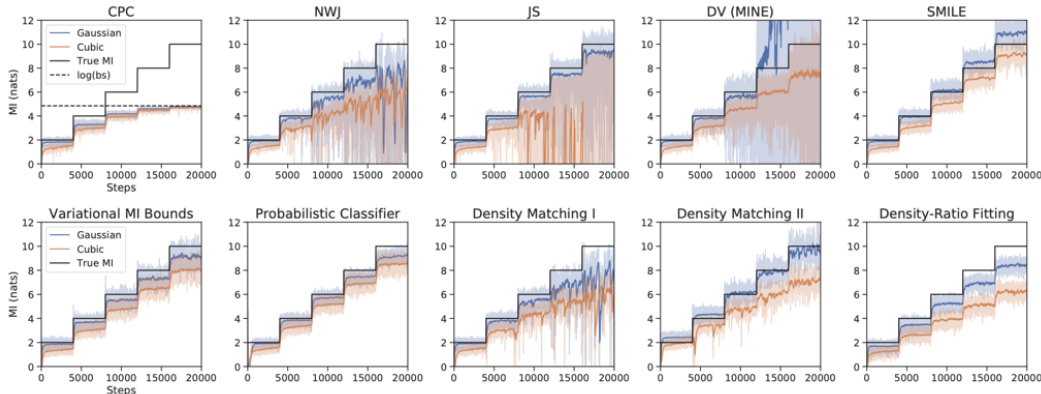

Figure 1: **Gaussian** and **Cubic** task for correlated Guassians with tractable ground truth MI. The upper row are the baselines and the lower row are our methods. Network, learning rate, optimizer, and batch size are fixed for all MI neural estimators. The only differences are the learning and inference objectives shown in Table 1.

**Baseline Models**  Instead of approximating MI by plugging-in the estimated PD or PMI, prior work focuses on establishing tractable and scalable bounds for MI [9, 36, 40, 41], in which the bounds can be computed via gradient descent over neural networks. Strong baselines include CPC [36], NWJ [9], JS [40], DV (MINE) [9], and SMILE [41]. To understand the differences, we separate MI neural estimation methods into two procedures: *learning* and *inference*. The learning step learns the parameters when estimating 1) point-wise dependency/ logarithm of point-wise dependency; or 2) MI lower bound. The inference step considers the parameters from the learning step and infers value for 1) MI itself; or 2) a lower bound of MI. We summarize different approaches in Table 1. For completeness, one may see Supplementary for more details about these bounds.

**Benchmarking on Correlated Gaussians**  To evaluate the performance between different MI neural estimators, we consider the standard tasks on correlated Gaussians [9, 40, 41]. In particular, we draw $(x, y)$ from two 20-dimensional Gaussians with correlation $\rho$, which is referred as **Gaussian** task. Then, we apply a cubic transformation on $y$ so that $y \mapsto y^3$, which is referred to as **Cubic** task. These two tasks have tractable ground truth MI $= -10 \log (1 - \rho^2)$. We train all models for $20,000$ iterations, starting from MI $= 2$ and increasing it by 2 per $4,000$ iterations. We fix the network, learning rate, optimizer, and batch size across all the estimators for a fair comparison. The only differences are the objectives considered in the learning and inference in MI estimation (shown in Table 1).

**Results & Discussions**  We present the results in Figure 1 and leave more training details in Supplementary. In the following, we discuss bias-variance trade-offs for different approaches. We first discuss general observations. Most of the estimators have both larger bias and variance with larger ground truth MI. The only exception is CPC [36], where its value is upper bounded by $\log (\texttt{batch\_size})$ [40]. The bias is also larger in **Gaussian** task than in **Cubic** task except for DV [9]. Next, we discuss the differences among estimators in detail. CPC [36] has the smallest variance, yet it is highly biased. Although having larger variance than CPC, SMILE [41]/ Variational MI Bounds/ Probabilistic Classifier/ Density Matching I & II/ Density-Ratio Fitting approaches have a much lower bias. Among them, Probabilistic Classifier and Density-Ratio Fitting approaches have the smallest variance. NWJ [9]/ JS [40]/ DV [9], whereas, have both large variance and bias. Note that JS [40] has larger variance is because using $I_{\text{NWJ}}$ objective during inference. To sum up, we see

that the plug-in MI estimators enjoy smaller variance and bias when comparing to most of the lower bound methods.

**Theoretical Analysis**    In Eq. (7), we present a high-level intuition that a good estimation of the PD function $\hat{r}_\theta(x, y)$ could be used to estimate the mutual information. In what follows, we present a formal justification for this argument. To begin with, let $P_{X,Y}^{(n)}$ denote the empirical distribution of the ground-truth joint distribution $P_{X,Y}$ estimated from $n$ samples drawn uniformly at random from $P_{X,Y}$. Then our estimator of the mutual information is given by $\widehat{I}_\theta^{(n)}(X;Y) := \mathbb{E}_{P_{X,Y}^{(n)}}[\log \hat{r}_\theta(x, y)]$.

At a high level, our arguments contain two parts. In the first part, we show that w.h.p. (with high probability) $\widehat{I}_\theta^{(n)}(X;Y)$ is close to $\mathbb{E}_{P_{X,Y}}[\log \hat{r}_\theta(x, y)]$. In the second part, we apply the universal approximation lemma of neural networks [22] to show that there exists $\hat{r}_\theta(\cdot, \cdot)$ that is close to $r(\cdot, \cdot)$. Formally, let $\mathcal{F} := \{\hat{r}_\theta : \theta \in \Theta \subseteq \mathbb{R}^d\}$ be the set of neural networks where the parameter $\theta$ is a $d$-dimensional vector. Throughout the analysis, we assume the following assumptions:

**Assumption 1** (Boundedness of the density ratio). There exist universal constants $C_l \leq C_u$ such that $\forall \hat{r}_\theta \in \mathcal{F}$ and $\forall x, y, C_l \leq \log \hat{r}_\theta(x, y) \leq C_u$.

**Assumption 2** (log-smoothness of the density ratio). There exists $\rho > 0$ such that for $\forall x, y$ and $\forall \theta_1, \theta_2 \in \Theta, |\log \hat{r}_{\theta_1}(x, y) - \log \hat{r}_{\theta_2}(x, y)| \leq \rho \cdot \|\theta_1 - \theta_2\|$.

Assumption 1 basically asks the output of a neural net to be bounded and Assumption 2 says that for any given input pair, the output of the network should only change slightly if we just slightly perturb the network weights. Both assumptions are mostly verified in practical networks. Based on these two assumptions, the following lemma is adapted from Bartlett [7] that bounds the rate of uniform convergence of a function class in terms of its covering number. The original lemma is based on the $L_\infty$ norm of the function class; whereas the following one, we use the $L_2$ norm on $\Theta$.

**Lemma 1.** (estimation). Let $\varepsilon > 0$ and $\mathcal{N}(\Theta, \varepsilon)$ be the covering number of $\Theta$ with radius $\varepsilon$ under $L_2$ norm. Let $P_{X,Y}$ be any distribution where $S = \{x_i, y_i\}_{i=1}^n$ are sampled from and define $M := C_u - C_l$, then

$$\Pr_S \left( \sup_{\hat{r}_\theta \in \mathcal{F}} \left| \widehat{I}_\theta^{(n)}(X;Y) - \mathbb{E}_{P_{X,Y}}[\log \hat{r}_\theta(x, y)] \right| \geq \varepsilon \right) \leq 2\mathcal{N}(\Theta, \varepsilon/4\rho) \exp\left( -\frac{n\varepsilon^2}{2M^2} \right). \quad (8)$$

Next lemma is derived from [22], which shows that neural networks are universal approximators:

**Lemma 2** (Hornik et al. [22], approximation). Let $\varepsilon > 0$. There exists $d \in \mathbb{N}$ and a family of neural networks $\mathcal{F} := \{\hat{r}_\theta : \theta \in \Theta \subseteq \mathbb{R}^d\}$ where $\Theta$ is compact, such that $\inf_{\hat{r}_\theta \in \mathcal{F}} \left| \mathbb{E}_{P_{X,Y}}[\log \hat{r}_\theta(x, y)] - I(X;Y) \right| \leq \varepsilon$.

Combining both lemmas, we are ready to state the following main result:

**Theorem 1.** Let $0 < \delta < 1$. There exists $d \in \mathbb{N}$ and a family of neural networks $\mathcal{F} := \{\hat{r}_\theta : \theta \in \Theta \subseteq \mathbb{R}^d\}$ where $\Theta$ is compact, so that $\exists \theta^* \in \Theta$, with probability at least $1 - \delta$ over the draw of $S = \{x_i, y_i\}_{i=1}^n \sim P_{X,Y}^{\otimes n}$,

$$\left| \widehat{I}_{\theta^*}^{(n)}(X;Y) - I(X;Y) \right| \leq O\left( \sqrt{\frac{d + \log(1/\delta)}{n}} \right). \quad (9)$$

It is worth pointing out that the above theorem is a theorem of existence, but *not* a constructive theorem, meaning that it does not give an estimator explicitly. To sum up, it shows that there exists a neural network $\theta^*$ such that, w.h.p., $\widehat{I}_{\theta^*}^{(n)}(X;Y)$ can approximate $I(X;Y)$ with $n$ samples at a rate of $O(1/\sqrt{n})$.

## 5    Application II: Self-supervised Representation Learning

Self-supervised representation learning aims at extracting task-relevant information without access to label or downstream signals. Among different self-supervised representation learning techniques,

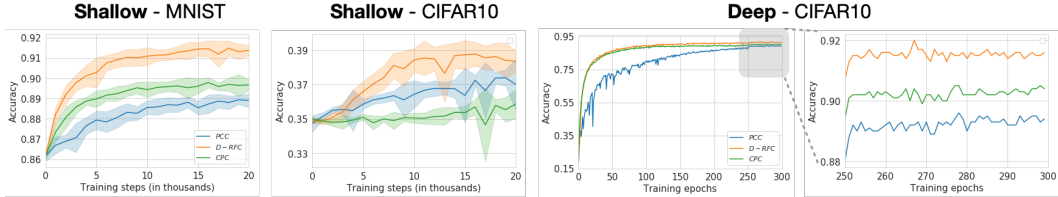

Figure 2: **Shallow** [48] and **Deep** [5] task for self-supervised visual representation learning using *downstream linear evaluation protocol*. We compare the presented Probabilistic Classifier Coding (PCC) and Density-Ratio Fitting Coding (D-RFC) with baseline Contrastive Predictive Coding (CPC). Network, learning rate, optimizer, and batch size are fixed for all the methods. The only differences are the learning objectives.

*contrastive learning* may be the most popular one with empirical [2, 3, 5, 13, 19–21, 23, 27, 36, 37, 45] and theoretical [4, 47] support. The core of contrastive learning is having the representations sampled from similar pairs be differentiated from random pairs. In other words, we hope that the representations learned from the similar pairs have higher point-wise dependency than the random pairs. Let $v_1/v_2$ denote two different views for the same data, $v_2'$ represent a view from a different data, and $F/G$ be two mapping functions from data to representations. In short, contrastive learning objective learns $F/G$ such that $r(F(v_1), G(v_2))$ is much larger than $r(F(v_1), G(v_2'))$.

**Connection between Contrastive Learning and PD**  Our goal is to show that our learning objectives resemble contrastive learning. We first take the *Probabilistic Classifier* approach as an example and incorporate the learning of $F/G$, which we name it as *Probabilistic Classifier Coding* (PCC):

$$\sup_{F,G} \sup_{\theta \in \Theta} \mathbb{E}_{P_{\mathcal{V}_1, \mathcal{V}_2}}[\log \hat{p}_\theta(c=1|(F(v_1), G(v_2)))] + \mathbb{E}_{P_{\mathcal{V}_1} P_{\mathcal{V}_2}}[\log \left(1 - \hat{p}_\theta(c=1|(F(v_1), G(v_2')))\right)], \quad (10)$$

which aims at learning $F/G$ to better classify (i.e., differentiate) between similar or random data pairs. Next, we consider the *Density-Ratio Fitting* approach, which we refer to the objective as *Density-Ratio Fitting Coding* (D-RFC):

$$\sup_{F,G} \sup_{\theta \in \Theta} \mathbb{E}_{P_{\mathcal{V}_1, \mathcal{V}_2}}[\hat{r}_\theta(F(v_1), G(v_2))] - \frac{1}{2}\mathbb{E}_{P_{\mathcal{V}_1} P_{\mathcal{V}_2}}[\hat{r}_\theta^2(F(v_1), G(v_2'))], \quad (11)$$

which aims at learning $F/G$ to maximize $\hat{r}_\theta(F(v_1), G(v_2))$ and minimize $\hat{r}_\theta(F(v_1), G(v_2'))$. We leave the discussion for the adaptations of Variational MI Bounds, Density Matching I ,and Density Matching II in Supplementary.

**Baseline Model**  The most adopted contrastive representation learning objective is Contrastive Predictive Coding (CPC) [36]:

$$\sup_{F,G} \sup_{\theta \in \Theta} \mathbb{E}_{(v_1^1, v_2^1) \sim P_{\mathcal{V}_1, \mathcal{V}_2}, \cdots (v_1^n, v_2^n) \sim P_{\mathcal{V}_1, \mathcal{V}_2}} \left[\frac{1}{n} \sum_{i=1}^n \log \frac{e^{\hat{c}_\theta(F(v_1^i), G(v_2^i))}}{\frac{1}{n} \sum_{j=1}^n e^{\hat{c}_\theta(F(v_1^i), G(v_2^j))}}\right],$$

where $\{v_1^i, v_2^i\}_{i=1}^n$ are independently and identically sampled from $P_{\mathcal{V}_1, \mathcal{V}_2}$. $\hat{c}_\theta(\cdot)$ is a function that takes the representations learned from the data pairs and returns a scalar.

**Experimental Setup**  We compare our proposed approaches with CPC [36] on two tasks [5, 48]. Due to the fact that the performance of the self-supervisedly learned representations strongly depends on the choice of feature extractor architectures and the parametrization of the employed MI estimators [48]. For a fair comparison, we fix the network, learning rate, optimizer, and batch size when comparing between different objectives. In the first set of experiments, we choose a relatively shallow network as suggested by Tschannen *et al.* [48], performing self-supervised learning experiments on MNIST [30] and CIFAR10 [29]. We report the average and standard deviations from 10 random trials. This task is referred to as **shallow** experiment. In the second set of experiments, we choose a relatively deep network as suggested by Bachman *et al.* [5], performing experiments on CIFAR10. This task is referred to as **deep** experiment. Both the **shallow** and **deep** tasks perform representation learning without access to the label information, and then the performance is evaluated by *downstream linear evaluation protocol* [5, 20, 21, 26, 36, 45, 48]. Specifically, a linear classifier is trained from the self-supervisedly learned (fixed) representation to the labels on the training set. We present the results with convergence in Figure 2. One may see Supplementary for more details.

Table 2: Cross-modal Retrieval task with unsupervised word features across acoustic and textual modalities. *Probabilistic Classifier* approach is used to estimate PD between the audio and textual features of a given word. The estimator is trained on the training split. We report the $1:5$ matching results from audio to textual features on the test split, where we obtain 96.24% top-1 retrieval accuracy.

| Correct Audio-Textual Retrieval Examples (Top-1 Accuracy: 96.24%) | | | | |
|---|---|---|---|---|
| Audio Feature | Textual Features (Ranked by logarithm of point-wise dependency) | | | |
| depths | **depths (15.22)** | mildewed (-58.62) | lugged (-92.24) | alison (-108.02) | raffleshurst (-161.74) |
| receptacle | **receptacle (1.32)** | bloated (-15.41) | recreate (-39.77) | sting (-90.51) | pity (-104.44) |
| frontiers | **frontiers (3.36)** | institution (-31.01) | laterally (-54.17) | pretends (-105.11) | vibrating (-124.88) |

| Incorrect Audio-Textual Retrieval Examples | | | | |
|---|---|---|---|---|
| Audio Feature | Textual Features (Ranked by logarithm of point-wise dependency) | | | |
| cos | tortoise (-2.33) | **cos (-10.72)** | tickling (-12.53) | undressed (-18.11) | cromwell's (-44.31) |
| elbowing | itinerary (-6.51) | **elbowing (-8.22)** | swims (-12.98) | rigid (-24.14) | integrity (-39.76) |
| alma's | roughness (-3.11) | **alma's (-3.67)** | montreal (-11.81) | tuneful (-12.22) | levant (-18.26) |

**Results & Discussions**  Prior approaches [37, 40, 41, 48] contend that a valid MI lower bound or an objective with better MI estimation may not result in better representations. We have a similar observation that D-RFC performs the best (when comparing to CPC and PCC) while it is neither a lower bound of MI nor the best objective of MI estimation. Next, we see an inconsistent trend when comparing PCC to CPC. In the Shallow task on CIFAR10, PCC performs better than CPC, while it performs worse on the other experiments. To sum up, we show our PD estimation objectives can be used for self-supervised representation learning, which is either at par or better than prior approaches.

## 6 Application III: Cross-modal Learning

In this section, we discuss the usage of point-wise dependency (PD) estimation for data containing information across modalities - audio and text.

**Experimental Setup - Cross-modal Retrieval**  We instantiate the discussion using unsupervised word features[2] which are learned from text corpora (i.e., Word2Vec [33] method) and human speech (i.e., Speech2Vec [15] method). In particular, in this dataset, a word feature has two distinct features: audio and textual feature. We denote $\mathcal{X}$ as the audio sample space and $\mathcal{Y}$ as the textual sample space. Since our goal is not comparing between different approaches but presenting the usage of PD estimation for cross-modal learning, we select only one approach *Probabilistic Classifier* as our objective for estimating PD. Note that we report the logarithm of PD, which is PMI in the results. One may refer to Supplementary for more details on training and datasets.

By definition, given an audio feature $x$ and a textual feature $y$, their point-wise dependency $r(x,y)$ measures their statistical dependency. For example, if $x_1$ and $y_1$ are the features for the same word, and $y_2$ is the feature for another word, then $r(x_1, y_1) > r(x_1, y_2)$ (in most cases). As a consequence, we can train PD estimators using the training split, and computing PD values for cross-modal retrieval on the test split.

**Results & Discussions**  In Table 2, we report the results on $1:5$ matching[3] from audio to textual features. First, we obtain 96.24% top-1 retrieval accuracy using PD estimation (with *Probabilistic Classifier* approach). Another approach such as *Density-Ratio Fitting* obtains 92.26% top-1 retrieval accuracy. Then, we study the success and failure retrieval cases. The success examples show the highest statistical dependency (i.e., the highest PMI) between the audio and textual features of the same word. The failure examples, on the contrary, (all of them) have the second-highest PMI between the audio and textual features of the same word. Last, we observe that only the correctly retrieved cross-modal features have positive PMI values, which suggest two features are statistically dependent.

As a summary, PD acts as a statistical dependency measurement, and we show its estimation can be generalized from training to test split for cross-modal retrieval.

## 7 Conclusion

In contrast to mutual information, which is an aggregate statistic of the dependency between two random variables, this paper contributes to present methods for estimating instance-level dependency. To overcome the curse of dimensionality in classical kernel-based approaches, we leverage the power of rich and flexible neural networks to model high-dimensional data. In particular, we first show that point-wise dependency is a natural product from optimizing mutual information variational bounds. Then, we further develop two point-wise dependency estimation approaches: Probabilistic Classifier and Density-Ratio Fitting that are free of optimizing mutual information variational bounds. A diversified set of experiments manifest the advantages of using our approaches. We believe this work sheds light on the advantages of estimating instance-level dependency between high-dimensional data, making a step forward towards improving unsupervised or cross-modal representation learning.

## Broader Impact

This paper presents methods for estimating point-wise dependency between high-dimensional data using neural networks. This work may benefit the applications that require understanding instance-level dependency. Take adversarial samples detection as an example: we can perform point-wise dependency estimation between data and label, and the ones with low point-wise dependency can be regarded as adversarial samples. We should also be aware of the malicious usage for our framework. For instance, people with bad intentions can use our framework to detect samples that have a high point-wise dependency with their of-interest private attributes. Then, these detected samples may be used for malicious purposes.

## Acknowledgements

This work was supported in part by the DARPA grants FA875018C0150 HR00111990016, NSF IIS1763562, NSF Awards #1750439 #1722822, National Institutes of Health, and Apple. We would also like to acknowledge NVIDIA's GPU support.

## Footnotes

*Work done at Carnegie Mellon University.

[2]The word features can be downloaded from https://github.com/iamyuanchung/speech2vec-pretrained-vectors.

[3]One trial contains an audio feature, its corresponding textual feature, and 4 randomly sampled textual features.

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
