[Supplementary Material]

# Supplementary for
# Neural Methods for Point-wise Dependency Estimation

**Yao-Hung Hubert Tsai**[1], **Han Zhao**[2]*,
**Makoto Yamada**[34], **Louis-Philippe Morency**[1], **Ruslan Salakhutdinov**[1]
[1]Carnegie Mellon University, [2]D.E. Shaw & Co., [3] Kyoto University, [4]RIKEN AIP

## 1  Optimization Objectives for Point-wise Dependency Neural Estimation

In this section, we shall show detailed derivations for the point-wise dependency estimation methods. Four approaches are discussed: *Variational Bounds of Mutual Information*, *Density Matching*, *Probabilistic Classifier*, and *Density-Ratio Fitting*. For convenience, we define $\Omega = \mathcal{X} \times \mathcal{Y}$. We have $P_{X,Y}$ and $P_X P_Y$ (can also be written as $P_X \otimes P_Y$) be the probability measures over $\sigma-$algebras over $\Omega$ with their probability densities being the Radon-Nikodym derivatives (i.e., $p(x,y) = dP_{X,Y}/d\mu$ and $p(x)p(y) = dP_X P_Y/d\mu$ with $\mu$ being the Lebesgue measure).

### 1.1  Method I: Variational Bounds of Mutual Information

Recent advances [5, 22] propose to estimate mutual information (MI) using neural network either from variational MI lower bounds (e.g., $I_{\text{NWJ}}$ [5] and $I_{\text{DV}}$ [5]) or a variational form of MI (e.g., $I_{\text{JS}}$ [22]). These estimators have the logarithm of point-wise dependency (PMI) as the intermediate product, which we will show in the following. We denote $\mathcal{M}$ be any class of functions $m : \Omega \to \mathbb{R}$.

**Proposition 1** ($I_{\text{NWJ}}$ and its neural estimation, restating Nguyen-Wainwright-Jordan bound [5, 18])**.**

$$I_{\text{NWJ}} := \sup_{m \in \mathcal{M}} \mathbb{E}_{P_{X,Y}}[m(x,y)] - e^{-1}\mathbb{E}_{P_X P_Y}[e^{m(x,y)}] = \sup_{\theta \in \Theta} \mathbb{E}_{P_{X,Y}}[\hat{f}_\theta(x,y)] - e^{-1}\mathbb{E}_{P_X P_Y}[e^{\hat{f}_\theta(x,y)}]$$

has the optimal function $m^*(x,y) = 1 + \log \frac{p(x,y)}{p(x)p(y)}$. And when $\Theta$ is large enough, the optimal $\hat{f}_\theta^*(x,y) = 1 + \log \frac{p(x,y)}{p(x)p(y)}$.

*Proof.* The second-order functional derivative of the objective is $-e^{-1} \cdot e^{m(x,y)} \cdot dP_X P_Y$, which is always negative. The negative second-order functional derivative implies the objective has a supreme value. Then, take the first-order functional derivative $\frac{\partial I_{\text{NWJ}}}{\partial m}$ and set it to zero:

$$dP_{X,Y} - e^{-1} \cdot e^{m(x,y)} \cdot dP_X P_Y = 0.$$

We then get optimal $m^*(x,y) = 1 + \log \frac{dP_{X,Y}}{dP_X P_Y} = 1 + \log \frac{p(x,y)}{p(x)p(y)}$. When $\Theta$ is large enough, by universal approximation theorem of neural networks [11], the approximation in Proposition 1 is tight, which means $\hat{f}_\theta^*(x,y) = m^*(x,y) = 1 + \log \frac{p(x,y)}{p(x)p(y)}$. ∎

**Proposition 2** ($I_{\text{DV}}$ and its neural estimation, restating Donsker-Varadhan bound [5, 8])**.**

$$I_{\text{DV}} := \sup_{m \in \mathcal{M}} \mathbb{E}_{P_{X,Y}}[m(x,y)] - \log\left(\mathbb{E}_{P_X P_Y}[e^{m(x,y)}]\right) = \sup_{\theta \in \Theta} \mathbb{E}_{P_{X,Y}}[\hat{f}_\theta(x,y)] - \log\left(\mathbb{E}_{P_X P_Y}[e^{\hat{f}_\theta(x,y)}]\right)$$

has optimal functions $m^*(x,y) = \log \frac{p(x,y)}{p(x)p(y)} + \text{Const.}$. And when $\Theta$ is large enough, the optimal $\hat{f}_\theta^*(x,y) = \log \frac{p(x,y)}{p(x)p(y)} + \text{Const.}$.

*Proof.* Let $\mathbf{1}.$ be an indicator function, and the second-order functional derivative of the objective is

$$-\frac{e^{m(x,y)}\cdot\mathbb{E}_{(x',y')\sim P_XP_Y}\left[e^{m(x',y')}\cdot\mathbf{1}_{(x',y')\neq(x,y)}\right]}{\left(\mathbb{E}_{P_XP_Y}[e^{m(x,y)}]\right)^2}\cdot dP_XP_Y,$$

which is always negative. The negative second-order functional derivative implies the objective has a supreme value. Then, take the first-order functional derivative $\frac{\partial I_{\mathrm{DV}}}{\partial m}$ and set it to zero:

$$dP_{X,Y}-\frac{e^{m(x,y)}}{\mathbb{E}_{P_XP_Y}[e^{m(x,y)}]}\cdot dP_XP_Y=0.$$

We then have $m^*(x,y)$ take the forms $m^*(x,y)=\log\frac{dP_{X,Y}}{dP_XP_Y}+\mathrm{Const.}=\log\frac{p(x,y)}{p(x)p(y)}+\mathrm{Const.}$. When $\Theta$ is large enough, by universal approximation theorem of neural networks [11], the approximation in Proposition 2 is tight, which means $\hat{f}^*_\theta(x,y)=m^*(x,y)=\log\frac{p(x,y)}{p(x)p(y)}+\mathrm{Const.}$. $\blacksquare$

**Proposition 3** ($I_{\mathrm{JS}}$ and its neural estimation, restating Jensen-Shannon bound with f-GAN objective [22])**.**

$$I_{\mathrm{JS}}:=\sup_{m\in\mathcal{M}}\mathbb{E}_{P_{X,Y}}\left[-\mathrm{softplus}\left(-m(x,y)\right)\right]-\mathbb{E}_{P_XP_Y}\left[\mathrm{softplus}\left(m(x,y)\right)\right]$$

$$=\sup_{\theta\in\Theta}\mathbb{E}_{P_{X,Y}}\left[-\mathrm{softplus}\left(-\hat{f}_\theta(x,y)\right)\right]-\mathbb{E}_{P_XP_Y}\left[\mathrm{softplus}\left(\hat{f}_\theta(x,y)\right)\right]$$

with softplus function being $\mathrm{softplus}\,(x)=\log\left(1+\exp\,(x)\right)$ and the optimal solution $m^*(x,y)=\log\frac{p(x,y)}{p(x)p(y)}$. And when $\Theta$ is large enough, the optimal $\hat{f}^*_\theta(x,y)=m^*(x,y)=\log\frac{p(x,y)}{p(x)p(y)}$.

*Proof.* The second-order functional derivative of the objective is

$$-\frac{1}{\left(1+e^{m(x,y)}\right)^2}\cdot e^{m(x,y)}\cdot dP_{X,Y}-\frac{1}{\left(1+e^{-m(x,y)}\right)^2}\cdot e^{-m(x,y)}\cdot dP_XP_Y,$$

which is always negative. The negative second-order functional derivative implies the objective has a supreme value. Then, take the first-order functional derivative $\frac{\partial I_{\mathrm{JS}}}{\partial m}$ and set it to zero:

$$\frac{1}{1+e^{-m(x,y)}}\cdot e^{-m(x,y)}\cdot dP_{X,Y}-\frac{1}{1+e^{m(x,y)}}\cdot e^{m(x,y)}\cdot dP_XP_Y=0.$$

We then get $m^*(x,y)=\log\frac{dP_{X,Y}}{dP_XP_Y}=\log\frac{p(x,y)}{p(x)p(y)}$. When $\Theta$ is large enough, by universal approximation theorem of neural networks [11], the approximation in Proposition 3 is tight, which means $\hat{f}^*_\theta(x,y)=m^*(x,y)=\log\frac{p(x,y)}{p(x)p(y)}$. $\blacksquare$

We see that either $I_{\mathrm{NWJ}}$ (Proposition 1) or $I_{\mathrm{JS}}$ (Proposition 3) gives us the optimal PMI estimation, while $I_{\mathrm{DV}}$ (Proposition 2) is less preferable since its optimal solution includes an arbitrary constant. In practice, we prefer $I_{\mathrm{JS}}$ over $I_{\mathrm{NWJ}}/I_{\mathrm{DV}}$ due to its better training stability [22].

## 1.2 Method II: Density Matching

This method considers to match the true joint density $p(x,y)$ and the estimated joint density via KL-divergence. We let the estimated joint probability be $P_{m,X,Y}$ with its joint density being $e^{m(x,y)}p(x)p(y)$, where $e^{m(x,y)}$ acts to ensure the estimated joint density is a valid probability density function. Hence, we let $m\in\mathcal{M}''$ with $\mathcal{M}''$ being 1) any class of functions $m:\Omega\to\mathbb{R}$; and 2) $\int e^{m(x,y)}\,dP_XP_Y=\mathbb{E}_{P_XP_Y}[e^{m(x,y)}]=1$.

**Proposition 4** (KL Loss in Density Matching and its neural estimation)**.**

$$L_{\mathrm{KL_{DM}}} := \sup_{m \in \mathcal{M}''} \mathbb{E}_{P_{X,Y}}[m(x,y)]$$

$$= \sup_{\theta \in \Theta} \mathbb{E}_{P_{X,Y}}[\hat{f}_\theta(x,y)] \text{ s.t. } \mathbb{E}_{P_X P_Y}[e^{\hat{f}_\theta(x,y)}] = 1$$

with the optimal $m^*(x,y) = \log \frac{p(x,y)}{p(x)p(y)}$. And when $\Theta$ is large enough, the optimal $\hat{f}_\theta^*(x,y) = \log \frac{p(x,y)}{p(x)p(y)}$.

*Proof.* First, we compute the KL-divergence:

$$L_{\mathrm{KL_{DM}}} = \inf_{m \in \mathcal{M}''} D_{\mathrm{KL}}(P_{X,Y} \parallel \hat{P}_{X,Y}) = \inf_{m \in \mathcal{M}''} H(P_{X,Y}) - \mathbb{E}_{P_{X,Y}}\Big[\log e^{m(x,y)} p(x)p(y)\Big]$$

$$= \inf_{m \in \mathcal{M}''} H(P_{X,Y}) - \mathbb{E}_{P_{X,Y}}\Big[\log p(x)p(y)\Big] - \mathbb{E}_{P_{X,Y}}\Big[m(x,y)\Big]$$

$$= \inf_{m \in \mathcal{M}''} I(X;Y) - \mathbb{E}_{P_{X,Y}}\Big[m(x,y)\Big] = \mathrm{Const.} + \sup_{m \in \mathcal{M}''} \mathbb{E}_{P_{X,Y}}\Big[m(x,y)\Big]$$

$$\Leftrightarrow \sup_{m \in \mathcal{M}} \mathbb{E}_{P_{X,Y}}[m(x,y)] \text{ s.t. } \mathbb{E}_{P_X P_Y}[e^{m(x,y)}] = 1.$$

Consider the following Lagrangian:

$$h(m, \lambda_1, \lambda_2) := \mathbb{E}_{P_{X,Y}}[m] - \lambda(\mathbb{E}_{P_X P_Y}[e^m] - 1),$$

where $\lambda \in \mathbb{R}$. Taking the functional derivative and setting it to be zero, we see

$$dP_{X,Y} - \lambda \cdot e^m \cdot dP_X dP_Y = 0.$$

To satisfy the constraint, we obtain

$$\mathbb{E}_{P_X P_Y}[e^m] = 1 \iff E_{P_X P_Y}\Big[\frac{1}{\lambda} \frac{dP_{X,Y}}{dP_X P_Y}\Big] = \frac{1}{\lambda} E_{P_X P_Y}\Big[\frac{dP_{X,Y}}{dP_X P_Y}\Big] = \frac{1}{\lambda} = 1 \iff \lambda = 1.$$

Plugging-in $\lambda = 1$, the optimal $m^*(x,y) = \log \frac{dP_{XY}}{dP_X P_Y} = \log \frac{p(x,y)}{p(x)p(y)}$. When $\Theta$ is large enough, by universal approximation theorem of neural networks [11], the approximation in Proposition 4 is tight, which means $\hat{f}_\theta^*(x,y) = m^*(x,y) = \log \frac{p(x,y)}{p(x)p(y)}$. ∎

The objective function in Proposition 4 is a constrained optimization problem, and we present two relaxed optimization objectives. The first one is Lagrange relaxation:

$$\sup_{\theta \in \Theta} \mathbb{E}_{P_{X,Y}}[\hat{f}_\theta(x,y)] - \lambda\Big(\mathbb{E}_{P_X P_Y}[e^{\hat{f}_\theta(x,y)}] - 1\Big)$$

with the optimal Lagrange coefficient $\lambda = 1$ (see proof for Proposition 4).

The second one is log barrier method:

$$\sup_{\theta \in \Theta} \mathbb{E}_{P_{X,Y}}[\hat{f}_\theta(x,y)] - \eta\Big(\log \mathbb{E}_{P_X P_Y}[e^{\hat{f}_\theta(x,y)}]\Big)^2,$$

where $\eta > 0$ is a hyper-parameter controlling the regularization term.

## 1.3 Method III: Probabilistic Classifier

This approach casts the PD estimation as the problem of estimating the 'class'-posterior probability. We use a Bernoulli random variable $C$ to classify the samples drawn from the joint density ($C = 1$ for $(x,y) \sim P_{X,Y}$) and the samples drawn from product of the marginal densities ($C = 0$ for $(x,y) \sim P_X P_Y$). In order to present our derivation, we define $H(\cdot)$ as the entropy and $H(\cdot, \cdot)$ as the cross entropy. Slightly abusing notation, in this subsection, we define $\Omega' = \mathcal{X} \times \mathcal{Y} \times \{0,1\}$ and $\mathcal{M}'$ is 1) any class of functions $m : \Omega' \to (0,1)$; and 2) $m(x,y,0) + m(x,y,1) = 1$ for any $x$ and $y$. Note that since $m(x,y,c)$ is always positive and $m(x,y,0) + m(x,y,1) = 1$ for any $x, y$, $m(x,y,c)$ is a proper probability mass function with respect to $C$ given any $x, y$. Consider the binary cross entropy loss:

**Proposition 5** (Binary Cross Entropy Loss in Probabilistic Classifier Method and its neural estimation).

$$L_{\mathrm{BCE}_{\mathrm{PC}}} := \sup_{m \in \mathcal{M}'} \mathbb{E}_{P_{X,Y}}[\log m(x,y,C=1)] + \mathbb{E}_{P_X P_Y}\left[\log\left(1 - m(x,y,C=1)\right)\right]$$

$$= \sup_{\theta \in \Theta} \mathbb{E}_{P_{X,Y}}[\log \hat{p}_\theta(C=1|(x,y))] + \mathbb{E}_{P_X P_Y}\left[\log\left(1 - \hat{p}_\theta(C=1|(x,y))\right)\right]$$

with the optimal $m^*(x,y,c) = p(c|(x,y))$. And when $\Theta$ is large enough, the optimal $\hat{p}_\theta^*(c|(x,y)) = p(c|(x,y))$.

*Proof.* We see

$$L_{\mathrm{BCE}_{\mathrm{PC}}} = \inf_{m \in \mathcal{M}'} \mathbb{E}_{P_{XY}}\left[H\left(P(C|(x,y)), m(x,y,C)\right)\right] + \mathbb{E}_{P_X P_Y}\left[H\left(P(C|(x,y)), m(x,y,C))\right)\right]$$

$$= \inf_{m \in \mathcal{M}'} \mathbb{E}_{P_{XY}}\left[H\left(P(C|(x,y))\right) + D_{\mathrm{KL}}(P(C|(x,y)) \parallel m(x,y,C))\right]$$

$$+ \mathbb{E}_{P_X P_Y}\left[H\left(P(C|(x,y))\right) + D_{\mathrm{KL}}(P(C|(x,y)) \parallel m(x,y,C))\right]$$

$$= \mathrm{Const.} + \inf_{m \in \mathcal{M}'} \mathbb{E}_{P_{XY}}\left[D_{\mathrm{KL}}(P(C|(x,y)) \parallel m(x,y,C))\right]$$

$$+ \mathbb{E}_{P_X P_Y}\left[D_{\mathrm{KL}}(P(C|(x,y)) \parallel m(x,y,C))\right]$$

$$= \mathrm{Const.} + \inf_{m \in \mathcal{M}'} \mathbb{E}_{P_{XY}}\left[\mathbb{E}_{P(C|(x,y))}[-\log m(x,y,c)]\right]$$

$$+ \mathbb{E}_{P_X P_Y}\left[\mathbb{E}_{P(C|(x,y))}[-\log m(x,y,c)]\right]$$

$$= \mathrm{Const.} + \inf_{m \in \mathcal{M}'} \mathbb{E}_{P_{XY}}[-\log m(x,y,C=1)] + \mathbb{E}_{P_X P_Y}[-\log m(x,y,C=0)]$$

$$\Leftrightarrow \sup_{m \in \mathcal{M}'} \mathbb{E}_{P_{X,Y}}[\log m(x,y,C=1)] + \mathbb{E}_{P_X P_Y}\left[\log\left(1 - m(x,y,C=1)\right)\right].$$

The optimal $m^*$ happens when $D_{\mathrm{KL}}(P(C|(x,y)) \parallel m^*(x,y,C)) = 0$ for any $(x,y)$, which implies $m^*(x,y,c) = p(c|(x,y))$. When $\Theta$ is large enough, by universal approximation theorem of neural networks [11], the approximation in Proposition 5 is tight, which means $\hat{p}_\theta^*(c|(x,y)) = m^*(x,y,c) = p(c|(x,y))$. ∎

The obtained estimated class-posterior classifier can be used for approximating point-wise dependency (PD):

$$\hat{r}_\theta(x,y) = \frac{n_{P_X P_Y}}{n_{P_{X,Y}}} \frac{\hat{p}_\theta(C=1|(x,y))}{\hat{p}_\theta(C=0|(x,y))} \text{ with } (x,y) \sim P_{X,Y} \text{ or } (x,y) \sim P_X P_Y.$$

## 1.4 Method IV: Density-Ratio Fitting

Let $\mathcal{M}$ be any class of functions $m : \Omega \to \mathbb{R}$. This approach considers to minimize the expected (in $\mathbb{E}_{P_X P_Y}[\cdot]$) least-square difference between the true PD $r(x,y)$ and the estimated PD $m(x,y)$:

**Proposition 6** (Least-Square Loss in Density-Ratio Fitting and its neural estimation).

$$L_{\mathrm{LS}_{\mathrm{D-RF}}} := \sup_{m \in \mathcal{M}} \mathbb{E}_{P_{X,Y}}[m(x,y)] - \frac{1}{2}\mathbb{E}_{P_X P_Y}[m^2(x,y)] = \sup_{\theta \in \Theta} \mathbb{E}_{P_{X,Y}}[\hat{r}_\theta(x,y)] - \frac{1}{2}\mathbb{E}_{P_X P_Y}[\hat{r}_\theta^2(x,y)]$$

with the optimal $m^*(x,y) = \frac{p(x,y)}{p(x)p(y)}$. And when $\Theta$ is larger enough, the optimal $\hat{r}_\theta^*(x,y) = \frac{p(x,y)}{p(x)p(y)}$.

*Proof.*

$$L_{\mathrm{LS_{D-RF}}} = \inf_{m \in \mathcal{M}} \mathbb{E}_{P_X P_Y}\big[\big(r(x,y) - m(x,y)\big)^2\big]$$

$$= \inf_{m \in \mathcal{M}} \mathbb{E}_{P_X P_Y}[r^2(x,y)] - 2\mathbb{E}_{P_X P_Y}[r(x,y)m(x,y)] + \mathbb{E}_{P_X P_Y}[m^2(x,y)]$$

$$= \mathrm{Const.} + \inf_{m \in \mathcal{M}} \, -2\mathbb{E}_{P_X P_Y}[r(x,y)m(x,y)] + \mathbb{E}_{P_X P_Y}[m^2(x,y)]$$

$$= \mathrm{Const.} + \inf_{m \in \mathcal{M}} \, -2\mathbb{E}_{P_{XY}}[m(x,y)] + \mathbb{E}_{P_X P_Y}[m^2(x,y)]$$

$$\Leftrightarrow \sup_{m \in \mathcal{M}} \mathbb{E}_{P_{XY}}[m(x,y)] - \frac{1}{2}\mathbb{E}_{P_X P_Y}[m^2(x,y)].$$

Take the first-order functional derivative and set it to zero:

$$dP_{XY} - m(x,y) \cdot dP_X P_Y = 0.$$

We then get $m^*(x,y) = \frac{dP_{X,Y}}{dP_X P_Y} = \frac{p(x,y)}{p(x)p(y)}$. When $\Theta$ is large enough, by universal approximation theorem of neural networks [11], the approximation in Proposition 6 is tight, which means $\hat{r}_\theta^*(x,y) = m^*(x,y) = \frac{p(x,y)}{p(x)p(y)}$. ∎

## 2 More on Mutual Information Neural Estimation

In this section, we present more analysis on estimating mutual information (MI) using neural networks. Before going into more details, we would like to 1) show $I_{\mathrm{NWJ}}$ and $I_{\mathrm{DV}}$ are MI lower bounds; and 2) present $I_{\mathrm{CPC}}$ [20] objective.

**Lemma 1** ($I_{\mathrm{NWJ}}$ as a MI lower bound)**.**

$$\forall \theta \in \Theta, \quad I(X;Y) \geq \mathbb{E}_{P_{X,Y}}[\hat{f}_\theta(x,y)] - e^{-1}\mathbb{E}_{P_X P_Y}[e^{\hat{f}_\theta(x,y)}].$$

Therefore,

$$I(X;Y) \geq I_{\mathrm{NWJ}} := \sup_{\theta \in \Theta} \mathbb{E}_{P_{X,Y}}[\hat{f}_\theta(x,y)] - e^{-1}\mathbb{E}_{P_X P_Y}[e^{\hat{f}_\theta(x,y)}].$$

*Proof.* In Proposition 1, we show the supreme value of $\mathbb{E}_{P_{X,Y}}[\hat{f}_\theta(x,y)] - e^{-1}\mathbb{E}_{P_X P_Y}[e^{\hat{f}_\theta(x,y)}]$ happens when $\hat{f}_\theta^*(x,y) = 1 + \log\frac{p(x,y)}{p(x)p(y)}$. Plugging-in $\hat{f}_\theta^*(x,y)$, we get

$$\mathbb{E}_{P_{X,Y}}[\hat{f}_\theta^*(x,y)] - e^{-1}\mathbb{E}_{P_X P_Y}[e^{\hat{f}_\theta^*(x,y)}] = \mathbb{E}_{P_{X,Y}}[1 + \log\frac{p(x,y)}{p(x)p(y)}] - e^{-1}\mathbb{E}_{P_X P_Y}[e^1 \cdot \frac{p(x,y)}{p(x)p(y)}]$$

$$= 1 + \mathbb{E}_{P_{X,Y}}[\log\frac{p(x,y)}{p(x)p(y)}] - e^{-1} \cdot e^1 \cdot \mathbb{E}_{P_X P_Y}[\frac{p(x,y)}{p(x)p(y)}] = 1 + I(X;Y) - 1 = I(X;Y). \qquad \blacksquare$$

**Lemma 2** ($I_{\mathrm{DV}}$ as a MI lower bound)**.**

$$\forall \theta \in \Theta, \quad I(X;Y) \geq \mathbb{E}_{P_{X,Y}}[\hat{f}_\theta(x,y)] - \log\Big(\mathbb{E}_{P_X P_Y}[e^{\hat{f}_\theta(x,y)}]\Big).$$

Therefore,

$$I(X;Y) \geq I_{\mathrm{DV}} := \sup_{\theta \in \Theta} \mathbb{E}_{P_{X,Y}}[\hat{f}_\theta(x,y)] - -\log\Big(\mathbb{E}_{P_X P_Y}[e^{\hat{f}_\theta(x,y)}]\Big).$$

*Proof.* In Proposition 2, we show the supreme value of $\mathbb{E}_{P_{X,Y}}[\hat{f}_\theta(x,y)] - \log\Big(\mathbb{E}_{P_X P_Y}[e^{\hat{f}_\theta(x,y)}]\Big)$ happens when $\hat{f}_\theta^*(x,y) = \mathrm{Const.} + \log\frac{p(x,y)}{p(x)p(y)}$. Plugging-in $\hat{f}_\theta^*(x,y)$, we get

$$\mathbb{E}_{P_{X,Y}}[\hat{f}_\theta^*(x,y)] - \log\Big(\mathbb{E}_{P_X P_Y}[e^{\hat{f}_\theta^*(x,y)}]\Big)$$

$$= \mathbb{E}_{P_{X,Y}}[\mathrm{Const.} + \log\frac{p(x,y)}{p(x)p(y)}] - \log\Big(\mathbb{E}_{P_X P_Y}[e^{\mathrm{Const.} + \log\frac{p(x,y)}{p(x)p(y)}}]\Big)$$

$$= \mathrm{Const.} + \mathbb{E}_{P_{X,Y}}[\log\frac{p(x,y)}{p(x)p(y)}] - \mathrm{Const.} \cdot \mathbb{E}_{P_X P_Y}[\frac{p(x,y)}{p(x)p(y)}] = I(X;Y).$$

$\blacksquare$

**Proposition 7** ($I_{\mathrm{CPC}}$, restating Contrastive Predictive Coding [20]). With $\hat{c}_\theta(x,y)$ representing a real-valued measureable function on $\mathcal{X} \times \mathcal{Y}$ which is parametrized by a neural network $\theta$,

$$L_{\mathrm{CPC}} := \sup_{\theta \in \Theta} \mathbb{E}_{(x_1,y_1) \sim P_{X,Y}, \cdots (x_n,y_n) \sim P_{X,Y}} \Big[ \frac{1}{n} \sum_{i=1}^{n} \log \frac{e^{\hat{c}_\theta(x_i,y_i)}}{\frac{1}{n} \sum_{j=1}^{n} e^{\hat{c}_\theta(x_i,y_j)}} \Big]$$

with an upper bound value $\log n$.

*Proof.*

$$L_{\mathrm{CPC}} = \sup_{\theta \in \Theta} \mathbb{E}_{(x_1,y_1) \sim P_{X,Y}, \cdots (x_n,y_n) \sim P_{X,Y}} \Big[ \frac{1}{n} \sum_{i=1}^{n} \log \frac{e^{\hat{c}_\theta(x_i,y_i)}}{\frac{1}{n} \sum_{j=1}^{n} e^{\hat{c}_\theta(x_i,y_j)}} \Big]$$

$$= \sup_{\theta \in \Theta} \mathbb{E}_{(x_1,y_1) \sim P_{X,Y}, \cdots (x_n,y_n) \sim P_{X,Y}} \Big[ \frac{1}{n} \sum_{i=1}^{n} \log \frac{e^{\hat{c}_\theta(x_i,y_i)}}{\sum_{j=1}^{n} e^{\hat{c}_\theta(x_i,y_j)}} \Big] + \log n$$

$$\leq \sup_{\theta \in \Theta} \mathbb{E}_{(x_1,y_1) \sim P_{X,Y}, \cdots (x_n,y_n) \sim P_{X,Y}} \Big[ \frac{1}{n} \sum_{i=1}^{n} \log \frac{e^{\hat{c}_\theta(x_i,y_i)}}{e^{\hat{c}_\theta(x_i,y_i)}} \Big] + \log n$$

$$= \sup_{\theta \in \Theta} \mathbb{E}_{(x_1,y_1) \sim P_{X,Y}, \cdots (x_n,y_n) \sim P_{X,Y}} \Big[ \frac{1}{n} \sum_{i=1}^{n} \log 1 \Big] + \log n$$

$$= \log n.$$

∎

**Lemma 3** ($I_{\mathrm{CPC}}$ as a MI lower bound).

$$\forall \theta \in \Theta, \quad I(X;Y) \geq \mathbb{E}_{(x_1,y_1) \sim P_{X,Y}, \cdots (x_n,y_n) \sim P_{X,Y}} \Big[ \frac{1}{n} \sum_{i=1}^{n} \log \frac{e^{\hat{c}_\theta(x_i,y_i)}}{\frac{1}{n} \sum_{j=1}^{n} e^{\hat{c}_\theta(x_i,y_j)}} \Big].$$

Therefore,

$$I(X;Y) \geq I_{\mathrm{CPC}} := \sup_{\theta \in \Theta} \mathbb{E}_{(x_1,y_1) \sim P_{X,Y}, \cdots (x_n,y_n) \sim P_{X,Y}} \Big[ \frac{1}{n} \sum_{i=1}^{n} \log \frac{e^{\hat{c}_\theta(x_i,y_i)}}{\frac{1}{n} \sum_{j=1}^{n} e^{\hat{c}_\theta(x_i,y_j)}} \Big].$$

*Proof.* First, we use independent and identical random variables $X_1, X_2, \cdots, X_n$ and $Y_1, Y_2, \cdots, Y_n$ to represent the copies of $X$ and $Y$, where $(x_i, y_i) \sim P_{X_i,Y_i}$. Replacing the random variables in Lemma 1, we obtain

$$\forall \theta \in \Theta, \quad I(X_i; Y_{1:n}) \geq \mathbb{E}_{P_{X_i,Y_{1:n}}}[\hat{f}_\theta(x_i, y_{1:k})] - e^{-1} \mathbb{E}_{P_{X_i} P_{Y_{1:n}}}[e^{\hat{f}_\theta(x_i, y_{1:k})}].$$

Next, we define $\hat{f}_\theta(x_i, y_{1:k}) = 1 + \log \frac{e^{\hat{c}_\theta(x_i,y_i)}}{\frac{1}{n} \sum_{j=1}^{n} e^{\hat{c}_\theta(x_i,y_j)}}$ and get

$$\forall \theta \in \Theta, \quad I(X_i; Y_{1:n}) \geq 1 + \mathbb{E}_{P_{X_i,Y_{1:n}}} \Big[ \log \frac{e^{\hat{c}_\theta(x_i,y_i)}}{\frac{1}{n} \sum_{j=1}^{n} e^{\hat{c}_\theta(x_i,y_j)}} \Big] - \mathbb{E}_{P_{X_i} P_{Y_{1:n}}} \Big[ \frac{e^{\hat{c}_\theta(x_i,y_i)}}{\frac{1}{n} \sum_{j=1}^{n} e^{\hat{c}_\theta(x_i,y_j)}} \Big].$$

Since $Y_1, Y_2, \cdots, Y_n$ are independent and identical samples, $\mathbb{E}_{P_{X_i} P_{Y_{1:n}}} \big[ \frac{e^{\hat{c}_\theta(x_i,y_i)}}{\frac{1}{n} \sum_{j=1}^{n} e^{\hat{c}_\theta(x_i,y_j)}} \big] = \mathbb{E}_{P_{X_i} P_{Y_{1:n}}} \big[ \frac{e^{\hat{c}_\theta(x_i,y_{i'})}}{\frac{1}{n} \sum_{j=1}^{n} e^{\hat{c}_\theta(x_i,y_j)}} \big] \forall i' \in \{1, 2, \cdots, n\}$. Therefore, $\mathbb{E}_{P_{X_i} P_{Y_{1:n}}} \big[ \frac{e^{\hat{c}_\theta(x_i,y_i)}}{\frac{1}{n} \sum_{j=1}^{n} e^{\hat{c}_\theta(x_i,y_j)}} \big] = \frac{1}{n} \sum_{i'=1}^{n} \mathbb{E}_{P_{X_i} P_{Y_{1:n}}} \big[ \frac{e^{\hat{c}_\theta(x_i,y_{i'})}}{\frac{1}{n} \sum_{j=1}^{n} e^{\hat{c}_\theta(x_i,y_j)}} \big] = \mathbb{E}_{P_{X_i} P_{Y_{1:n}}} \big[ \frac{\frac{1}{n} \sum_{i'=1}^{n} e^{\hat{c}_\theta(x_i,y_{i'})}}{\frac{1}{n} \sum_{j=1}^{n} e^{\hat{c}_\theta(x_i,y_j)}} \big] = 1$. Plugging-in this result, we have

$$\forall \theta \in \Theta, \quad I(X_i; Y_{1:n}) \geq 1 + \mathbb{E}_{P_{X_i,Y_{1:n}}} \Big[ \log \frac{e^{\hat{c}_\theta(x_i,y_i)}}{\frac{1}{n} \sum_{j=1}^{n} e^{\hat{c}_\theta(x_i,y_j)}} \Big] - 1 = \mathbb{E}_{P_{X_i,Y_{1:n}}} \Big[ \log \frac{e^{\hat{c}_\theta(x_i,y_i)}}{\frac{1}{n} \sum_{j=1}^{n} e^{\hat{c}_\theta(x_i,y_j)}} \Big].$$

Note that $Y_{i'}$ is independent to $X_i$ when $i' \neq i$, and therefore $I(X_i; Y_{1:n}) = I(X_i; Y_i) = I(X;Y)$.

Bringing everything together, the original objective can be reformulated as

$$\mathbb{E}_{(x_1,y_1)\sim P_{X,Y},\cdots(x_n,y_n)\sim P_{X,Y}}\Big[\frac{1}{n}\sum_{i=1}^{n}\log\frac{e^{\hat{c}_\theta(x_i,y_i)}}{\frac{1}{n}\sum_{j=1}^{n}e^{\hat{c}_\theta(x_i,y_j)}}\Big]$$

$$=\mathbb{E}_{P_{X_{1:n},Y_{1:n}}}\Big[\frac{1}{n}\sum_{i=1}^{n}\log\frac{e^{\hat{c}_\theta(x_i,y_i)}}{\frac{1}{n}\sum_{j=1}^{n}e^{\hat{c}_\theta(x_i,y_j)}}\Big]=\frac{1}{n}\sum_{i=1}^{n}\mathbb{E}_{P_{X_i,Y_{1:n}}}\Big[\log\frac{e^{\hat{c}_\theta(x_i,y_i)}}{\frac{1}{n}\sum_{j=1}^{n}e^{\hat{c}_\theta(x_i,y_j)}}\Big]$$

$$\leq\frac{1}{n}\sum_{i=1}^{n}I(X_i;Y_{1:n})=\frac{1}{n}\sum_{i=1}^{n}I(X;Y)=I(X;Y).$$

∎

## 2.1 Learning/ Inference in MI Neural Estimation and Baselines

The MI neural estimation methods can be dissected into two procedures: *learning* and *inference*. The learning step learns the parameters when estimating 1) point-wise dependency (PD)/ logarithm of point-wise dependency (PMI); or 2) MI lower bound. The inference step considers the parameters from the learning step and infers value for 1) MI itself; or 2) a lower bound of MI. We summarize different approaches in Table 1 in the main text, and we discuss the baselines in this subsection. We present the comparisons between baselines and our methods in Table 1/ Figure 1 in the main text.

**CPC**  Oord *et al.* [20] presented **C**ontrastive **P**redictive **C**oding (**CPC**) as an unsupervised learning objective, which adopts $I_{\text{CPC}}$ (see Proposition 7) in both learning and inference stages. From Proposition 7 and Lemma 3, we conclude

$$I_{\text{CPC}}\leq\min\Big(\log n, I(X;Y)\Big).$$

Hence, the difference between $I_{\text{CPC}}$ and $I(X;Y)$ is large when $n$ is small. This fact implies a large bias when using $I_{\text{CPC}}$ to estimate MI. Nevertheless, empirical evidences [22, 23] showed that $I_{\text{CPC}}$ has low variance, which is also verified in our experiments.

**NWJ**  Belghazi *et al.* [5] presented to use neural networks to estimate **N**guyen-**W**ainwright-**J**ordan bound [5, 18] (**NWJ**) bound of MI, which adopts $I_{\text{NWJ}}$ (see Proposition 1) in both learning and inference stages. In Proposition 1 and Lemma 1, we show that when $\Theta$ is large enough, the supreme value of $I_{\text{NWJ}}$ is $I(X;Y)$. Hence, we can expect a smaller bias when comparing $I_{\text{NWJ}}$ to $I_{\text{CPC}}$. Song *et al.* [23] acknowledged the variance of an empirical $I_{\text{NWJ}}$ estimation is $\Omega(e^{I(X;Y)})$, suggesting a large variance when the true MI is large. We verify these facts in our experiments.

**DV (MINE)**  Belghazi *et al.* [5] presented to use neural networks to estimate **D**onsker-**V**aradhan bound [5, 18] (**DV**) bound of MI, which adopts $I_{\text{DV}}$ (see Proposition 2) in both learning and inference stages. The author also refers this MI estimation procedure as **M**utual **I**nformation **N**eural **E**stimation (**MINE**). In Proposition 2 and Lemma 2, we show that when $\Theta$ is large enough, the supreme value of $I_{\text{DV}}$ is $I(X;Y)$. Hence, we can expect a smaller bias when comparing $I_{\text{DV}}$ to $I_{\text{CPC}}$. Song *et al.* [23] acknowledged the limiting variance of an empirical $I_{\text{DV}}$ estimation is $\Omega(e^{I(X;Y)})$, which implies the variance is large when the true MI is large. We verify these facts in our experiments.

**JS**  Unlike **CPC**, **NWJ**, and **DV**, Poole *et al.* [22] presented to adopt different objectives in learning and inference stages for MI estimation. Precisely, the author uses Jensen-Shannon F-GAN [19] objective (see Proposition 3) to estimate PMI and then plugs in the PMI into $I_{\text{NWJ}}$ (see Proposition 1) for the inference. The author refers this MI estimation method as **JS** since it considers **J**ensen-**S**hannon divergence during learning. Unfortunately, this estimation method still considers $I_{\text{NWJ}}$ as its inference objective, and therefore the variance is still $\Omega(e^{I(X;Y)})$. Empirical results are shown in our experiments.

**SMILE**  To overcome the large variance issue in **NWJ**, **DV**, and **JS**, Song *et al.* [23] presented to use $I_{\text{JS}}$ (see Proposition 3) for estimating PMI and then plug in the PMI to a modified $I_{\text{DV}}$ (see Proposition 2). Specifically, the author clipped the value of $e^{\hat{f}_\theta(x,y)}$ in the second term of $I_{\text{DV}}$ to control the variance during the inference stage. Although the modification introduces some bias for MI estimation, it is empirically admitting a small variance, which we also find in our experiments.

## 2.2 Architecture Design in Experiments

We follow the same training and evaluation protocal for Correlated Gaussians experiments in prior work [22, 23]. We adopt the "concatenate critic" design [20, 22, 23] for our neural network parametrized function. The neural network parametrized functions are $\hat{c}_\theta$ in **CPC**, $\hat{f}_\theta$ in **NWJ/JS/DV/SMILE/**Variational MI Bounds/Density Matching I/Density Matchinig II, $\hat{r}_\theta$ in Density-Ratio Fitting, and $\hat{p}_\theta$ in Probabilistic Classifier. Take $\hat{c}_\theta$ as an example, the concatenate critic design admits $\hat{c}_\theta(x, y) = g_\theta([x, y])$ with $g_\theta$ being multiple-layer perceptrons. We consider $g_\theta$ to be 1-hidden-layer neural network with 512 neurons for each layer and ReLU function as the activation. The optimization considers batch size 128 and Adam optimizer [12] with learning rate 0.001. For a fair comparison, we fix everything except for the learning and inference objectives. Note that Probabilistic Classifier method applies sigmoid function to the outputs to ensure probabilistic outputs. We set $\eta = 1.0$ in Density Matching II.

**Reproducibility**   Please refer to our released code.

## 2.3 Theoretical Analysis

We restate the Assumptions in the main text:

**Assumption 1** (Boundedness of the density ratio; restating Assumption 1 in the main text). There exist universal constants $C_l \le C_u$ such that $\forall \hat{r}_\theta \in \mathcal{F}$ and $\forall x, y$, $C_l \le \log \hat{r}_\theta(x, y) \le C_u$.

**Assumption 2** (log-smoothness of the density ratio; restating Assumption 2 in the main text). There exists $\rho > 0$ such that for $\forall x, y$ and $\forall \theta_1, \theta_2 \in \Theta$, $|\log \hat{r}_{\theta_1}(x, y) - \log \hat{r}_{\theta_2}(x, y)| \le \rho \cdot \|\theta_1 - \theta_2\|$.

In what follows, we first prove the following lemma. The main idea is from Bartlett [4], while here we focus on the covering number of the parameter space $\Theta$ using $L_2$ norm.

**Lemma 4** (estimation; restating Lemma 1 in the main text). Let $\varepsilon > 0$ and $\mathcal{N}(\Theta, \varepsilon)$ be the covering number of $\Theta$ with radius $\varepsilon$ under $L_2$ norm. Let $P_{X,Y}$ be any distribution where $S = \{x_i, y_i\}_{i=1}^n$ are sampled from and define $M := C_u - C_l$, then

$$\Pr_S \left( \sup_{\hat{r}_\theta \in \mathcal{F}} \left| \widehat{I}_\theta^{(n)}(X; Y) - \mathbb{E}_{P_{X,Y}}[\log \hat{r}_\theta(x, y)] \right| \ge \varepsilon \right) \le 2\mathcal{N}(\Theta, \varepsilon/4\rho) \exp\left( -\frac{n\varepsilon^2}{2M^2} \right). \quad (1)$$

*Proof.* Define $l_S(\theta) := \widehat{I}_\theta^{(n)}(X; Y) - \mathbb{E}_{P_{X,Y}}[\log \hat{r}_\theta(x, y)]$. For $\theta_1, \theta_2 \in \Theta$, we first bound the difference $|l_S(\theta_1) - l_S(\theta_2)|$ in terms of the distance between $\theta_1$ and $\theta_2$. To do so, for any joint distribution $P$ over $X \times Y$, we first bound the following difference:

$$|\mathbb{E}_P[\log \hat{r}_{\theta_1}(x, y)] - \mathbb{E}_P[\log \hat{r}_{\theta_2}(x, y)]| \le \mathbb{E}_P[|\log \hat{r}_{\theta_1}(x, y) - \log \hat{r}_{\theta_2}(x, y)|]$$
$$\le \mathbb{E}_P[\rho \cdot \|\theta_1 - \theta_2\|_2]$$
$$= \rho \cdot \|\theta_1 - \theta_2\|_2,$$

where the first inequality is due to the triangle inequality and the second one is from Assumption 2. Next we bound $|l_S(\theta_1) - l_S(\theta_2)|$ by applying the above inequality twice:

$$|l_S(\theta_1) - l_S(\theta_2)| = \left| \left( \widehat{I}_{\theta_1}^{(n)}(X; Y) - \mathbb{E}_{P_{X,Y}}[\log \hat{r}_{\theta_1}(x, y)] \right) - \left( \widehat{I}_{\theta_2}^{(n)}(X; Y) - \mathbb{E}_{P_{X,Y}}[\log \hat{r}_{\theta_2}(x, y)] \right) \right|$$
$$\le \left| \widehat{I}_{\theta_1}^{(n)}(X; Y) - \widehat{I}_{\theta_2}^{(n)}(X; Y) \right| + \left| \mathbb{E}_{P_{X,Y}}[\log \hat{r}_{\theta_1}(x, y)] - \mathbb{E}_{P_{X,Y}}[\log \hat{r}_{\theta_2}(x, y)] \right|$$
$$\le \rho \cdot \|\theta_1 - \theta_2\| + \rho \cdot \|\theta_1 - \theta_2\|_2$$
$$= 2\rho \cdot \|\theta_1 - \theta_2\|.$$

Now we consider the covering of $\Theta$. Since $\Theta$ is compact, it admits a finite covering. To simplify the notation, let $T := \mathcal{N}(\Theta, \varepsilon/4\rho)$ and let $\cup_{k=1}^T \Theta_k$ be a finite cover of $\Theta$. Furthermore, assume $\theta_i \in \Theta_i$ be the center of the $L_2$ ball $\Theta_i$ with radius $\varepsilon/4\rho$. As a result, the following bound holds:

$$\Pr_S(\sup_{\hat{r}_\theta \in \mathcal{F}} |l_S(\theta)| \ge \varepsilon) = \Pr_S(\sup_{\theta \in \Theta} |l_S(\theta)| \ge \varepsilon)$$
$$\le \Pr_S(\cup_{k \in [T]} \sup_{\theta \in \Theta_k} |l_S(\theta)| \ge \varepsilon)$$
$$\le \sum_{k \in [T]} \Pr_S(\sup_{\theta \in \Theta_k} |l_S(\theta)| \ge \varepsilon).$$

The last inequality above is due to the union bound. Next, $\forall k \in [T]$, realize that the following inequality holds:
$$\Pr_S(\sup_{\theta \in \Theta_k} |l_S(\theta)| \geq \varepsilon) \leq \Pr_S(|l_S(\theta_k)| \geq \varepsilon/2).$$

To see this, note that the $L_2$ ball of $\Theta_k$ has radius $\varepsilon/4\rho$, hence $\sup_{\theta \in \Theta_k} |l_S(\theta) - l_S(\theta_k)| \leq 2\rho \cdot \varepsilon/4\rho = \varepsilon/2$, which yields:
$$\Pr_S(\sup_{\theta \in \Theta_k} |l_S(\theta)| \geq \varepsilon) \leq \Pr_S(\sup_{\theta \in \Theta_k} |l_S(\theta) - l_S(\theta_k)| + |l_S(\theta_k)| \geq \varepsilon)$$
$$\leq \Pr_S(|l_S(\theta_k)| \geq \varepsilon/2).$$

To proceed, it suffices if we could provide an upper bound for $\Pr_S(|l_S(\theta_k)| \geq \varepsilon/2)$. Now since $\log \hat{r}_{\theta_k}(x, y)$ is bounded for any pair of input $x, y$ by Assumption 1, it follows from the Hoeffding's inequality that

$$\Pr_S(|l_S(\theta_k)| \geq \varepsilon/2) = \Pr_S\left(\left|\widehat{I}_{\theta_k}^{(n)}(X;Y) - \mathbb{E}_{P_{X,Y}}[\log \hat{r}_{\theta_k}(x, y)]\right| \geq \varepsilon/2\right)$$
$$\leq 2 \exp\left(-\frac{n\varepsilon^2}{2M^2}\right).$$

Now, combine all the pieces together, we have:
$$\Pr_S\left(\sup_{\hat{r}_\theta \in \mathcal{F}} \left|\widehat{I}_\theta^{(n)}(X;Y) - \mathbb{E}_{P_{X,Y}}[\log \hat{r}_\theta(x, y)]\right| \geq \varepsilon\right) = \Pr_S(\sup_{\theta \in \Theta} |l_S(\theta)| \geq \varepsilon)$$
$$\leq \sum_{k \in [T]} \Pr_S(\sup_{\theta \in \Theta_k} |l_S(\theta)| \geq \varepsilon)$$
$$\leq \mathcal{N}(\Theta, \varepsilon/4\rho) \Pr_S(\sup_{\theta \in \Theta_k} |l_S(\theta)| \geq \varepsilon)$$
$$\leq \mathcal{N}(\Theta, \varepsilon/4\rho) \Pr_S(|l_S(\theta_k)| \geq \varepsilon/2)$$
$$\leq 2\mathcal{N}(\Theta, \varepsilon/4\rho) \exp\left(-\frac{n\varepsilon^2}{2M^2}\right). \qquad \blacksquare$$

We restate the Lemma 2 in the main text:

**Lemma 5** (Hornik et al. [11], approximation; restating Lemma 2 in the main text)**.** Let $\varepsilon > 0$. There exists $d \in \mathbb{N}$ and a family of neural networks $\mathcal{F} := \{\hat{r}_\theta : \theta \in \Theta \subseteq \mathbb{R}^d\}$ where $\Theta$ is compact, such that $\inf_{\hat{r}_\theta \in \mathcal{F}} \left|\mathbb{E}_{P_{X,Y}}[\log \hat{r}_\theta(x, y)] - I(X;Y)\right| \leq \varepsilon$.

Now, we are ready the present our theorem:

**Theorem 1.** Let $0 < \delta < 1$. There exists $d \in \mathbb{N}$ and a family of neural networks $\mathcal{F} := \{\hat{r}_\theta : \theta \in \Theta \subseteq \mathbb{R}^d\}$ where $\Theta$ is compact, so that $\exists \theta^* \in \Theta$, with probability at least $1 - \delta$ over the draw of $S = \{x_i, y_i\}_{i=1}^n \sim P_{X,Y}^{\otimes n}$,

$$\left|\widehat{I}_{\theta^*}^{(n)}(X;Y) - I(X;Y)\right| \leq O\left(\sqrt{\frac{d + \log(1/\delta)}{n}}\right). \tag{2}$$

*Proof.* This theorem simply follows a combination of Lemma 4 and Lemma 5. First, by Lemma 5, for $\varepsilon > 0$, there exists $d \in \mathbb{N}$ and a family of neural networks $\mathcal{F} := \{\hat{r}_\theta : \theta \in \Theta \subseteq \mathbb{R}^d\}$ where $\Theta$ is compact, such that there $\exists \theta^* \in \Theta$,

$$\left|\mathbb{E}_{P_{X,Y}}[\log \hat{r}_{\theta^*}(x, y)] - I(X;Y)\right| \leq \frac{\varepsilon}{2}.$$

Next, we perform analysis on the estimation error $\left|\widehat{I}_{\theta^*}^{(n)}(X;Y) - \mathbb{E}_{P_{X,Y}}[\log \hat{r}_{\theta^*}(x, y)]\right| \leq \frac{\varepsilon}{2}$. Applying Lemma 4 with the fact [2] that for $\Theta \subseteq \mathbb{R}^d$, $\log \mathcal{N}(\Theta, \varepsilon/4\rho) = O(d \log(\rho/\varepsilon))$, we can solve for $\varepsilon$ in terms of the given $\delta$. It suffices for us to find $\varepsilon \to \frac{\varepsilon}{2}$ such that:

$$2\mathcal{N}(\Theta, \varepsilon/8\rho) \exp\left(-\frac{n\varepsilon^2}{8M^2}\right) \leq \delta,$$

which is equivalent to finding $\varepsilon$ such that the following inequality holds:

$$c \cdot d \log \frac{\varepsilon}{8\rho} + \frac{n\varepsilon^2}{8M^2} \geq \log \frac{2}{\delta},$$

where $c$ is a universal constant that is independent of $d$. Now, using the inequality $\log(x) \leq x - 1$, it suffices for us to find $\varepsilon$ such that

$$c \cdot d \left( \frac{\varepsilon}{8\rho} - 1 \right) + \frac{n\varepsilon^2}{8M^2} \geq c \cdot d \log \frac{\varepsilon}{8\rho} + \frac{n\varepsilon^2}{8M^2} \geq \log \frac{2}{\delta},$$

which is in turn equivalent to solving:

$$\varepsilon^2 + c'\varepsilon \geq \left( \log \frac{2}{\delta} + cd \right) \cdot \frac{8M^2}{n},$$

where $c' = c'(c, d, \rho, n, M)$. Nevertheless, in order for the above inequality to hold, it suffices if we choose

$$\varepsilon = O\left( \sqrt{\frac{d + \log(1/\delta)}{n}} \right).$$

The final step is to combine the above two inequalities together:

$$\left| \widehat{I}_{\theta^*}^{(n)}(X;Y) - I(X;Y) \right| \leq \left| \widehat{I}_{\theta^*}^{(n)}(X;Y) - \mathbb{E}_{P_{X,Y}}[\log \hat{r}_{\theta^*}(x,y)] \right| + \left| \mathbb{E}_{P_{X,Y}}[\log \hat{r}_{\theta^*}(x,y)] - I(X;Y) \right|$$

$$\leq \frac{\varepsilon}{2} + \frac{\varepsilon}{2} = O\left( \sqrt{\frac{d + \log(1/\delta)}{n}} \right). \qquad \blacksquare$$

# 3 More on Self-supervised Representation Learning

In the main text, we have shown how we adapt the proposed point-wise dependency estimation approaches (Probabilistic Classifier and Density-Ratio Fitting) to contrastive learning objectives (Probabilistic Classifier Coding and Density-Ratio Fitting Coding) for self-supervised representation learning. Following the adaptation, it is straightforward to define new contrastive learning objectives that are inspired by other presented approaches such as Variational MI Bounds, Density Matching I ,and Density Matching II. Nevertheless, instead of presenting new objectives, we would like to discuss 1) the connection between Probabilistic Classifier and Variational MI Bounds; 2) the connection between Density Matchinig I/II and $I_{\mathrm{NWJ}}$ (see Proposition 1); and 3) the potential limitations of the new objectives. Next, we will discuss the baseline method Contrastive Predictive Coding (CPC). Last, we present the experimental details.

## 3.1 Connection between Probabilistic Classifier and Variational MI Bounds

Proposition 5 states that the Probabilistic Classifier approach admits a classification task to differentiate the pairs sampled from a joint distribution or the product of marginal distribution. This classification task minimizes the binary cross entropy loss, which is highly optimized and stabilized in popular optimization packages such as PyTorch [21] and TensorFlow [1] (e.g., log-sum-exp trick for numerical stability). Note that, if we let $\hat{p}_\theta = \mathrm{sigmoid}\left(l_\theta\right)$ with $l_\theta$ being the logits model, then reformulating Probabilistic Classifier to optimizing $l_\theta$ leads to the same objective as $I_{\mathrm{JS}}$ (see Proposition 3), which is the learning objective of *Variational MI Bounds* method. Although being the same objective as the Probabilistic Classifier approach, $I_{\mathrm{JS}}$ may encounter a relatively higher training instability (unless a particular take-care on its numerical instability). As pointed out by Tschannen *et al.* [25], contrastive learning approaches with higher variance may result in a lower down-stream task performance, which accords with our empirical observation.

## 3.2 Connection between Density Matching I/II and $I_{\mathrm{NWJ}}$

Density Matching I/II approaches are derived from the KL loss between the true joint density and estimated joint density ($L_{\mathrm{KL_{DM}}}$ in Proposition 4). Specifically, Density Matching I is a Lagrange relaxation of $L_{\mathrm{KL_{DM}}}$. If we change $\hat{f}_\theta + 1 = \hat{f}'_\theta$ in Density Matching I approach, then reformulating

our objective to optimizing $\hat{f}'_\theta$ leads to the same objective as $I_{\mathrm{NWJ}}$ (see Proposition 1). Song *et al.* [23] acknowledged the variance of an empirical $I_{\mathrm{NWJ}}$ estimation is $\Omega(e^{I(X;Y)})$, and hence the variance is large unless $I(X;Y)$ is small. Having the same conclusion in Sec 3.1, our empirical observation finds Density Matching I/II lead to worsened representation as comparing to other contrastive learning objectives.

## 3.3 Contrastive Predictive Coding (CPC) for Contrastive Representation Learning

Contrastive Predictive Coding (CPC) [20] adapts $I_{\mathrm{CPC}}$ (see Proposition 7) to a contrastive representation learning objective:

$$\sup_{F,G} \sup_{\theta \in \Theta} \mathbb{E}_{(v_1^1,v_2^1) \sim P_{\mathcal{V}_1,\mathcal{V}_2}, \cdots (v_1^n,v_2^n) \sim P_{\mathcal{V}_1,\mathcal{V}_2}} \Big[ \frac{1}{n} \sum_{i=1}^{n} \log \frac{e^{\hat{c}_\theta(F(v_1^i),G(v_2^i))}}{\frac{1}{n} \sum_{j=1}^{n} e^{\hat{c}_\theta(F(v_1^i),G(v_2^j))}} \Big],$$

where $\{v_1^i, v_2^i\}_{i=1}^n$ are independently and identically sampled from $P_{\mathcal{V}_1,\mathcal{V}_2}$. $\hat{c}_\theta(\cdot)$ is a function that takes the representations learned from the data pairs and returns a scalar.

## 3.4 Experiments Details

**Datasets**   We adopt MNIST [15] and CIFAR10 [14] as the datasets in our experiments. MNIST contains $60,000$ training and $10,000$ test examples. Each example is a grey-scale digit image ($0 \sim 9$) with size $28 \times 28$. CIFAR10 contains $50,000$ training and $10,000$ test examples. Each example is a $32 \times 32$ colour image from 10 mutual exclusive classes: {airplane, automobile, bird, cat, deer, dog, frog, horse, ship, truck}.

**Pre-training and Fine-tuning**   Our self-supervised learning experiments contain two stages: *pre-training* and *fine-tuning*. In pre-training stage, we learn representation from the training samples using contrastive learning objectives (e.g., Probabilistic Classifier Coding (PCC), Density-Ratio Fitting Coding (D-RFC), and Contrastive Predictive Coding (CPC) [20]). View 1 ($\mathcal{V}_1$) and 2 ($\mathcal{V}_2$) are generated by augmenting the input with different transformations. For example, given an input, $v_1$ can be the 15-degree-rotated one and $v_2$ can be the horizontally flipped one. For **shallow** experiment, we consider the same data augmentations adopted in Tschannen [25]; for **deep** experiment, we consider the same data augmentations adopted in Bachman [3]. In fine-tuning stage, the network in the pre-training stage is fixed; we train only the classifier for minimizing classification loss from the representations. We follow linear evaluation protocol [3, 9, 10, 13, 20, 24, 25] such that the classifier is a linear layer. After the pre-training and fine-tuning stages, we evaluate the performance of the model on the test samples.

**Architectures**   To clearly understand how contrastive learning objectives affect the down-stream performance, we fix the network, learnnig rate, optimizer, and batch size across different objectives. To be more precise, we stick to the official implementations by Tschannen *et al.* [25] (for **shallow** experiment) and Bachman *et al.* [3] (for **deep** experiment). The only change is the contrastive learning objective, which is the loss in the pre-training stage for self-supervised learning experiments.

**Reproducibility**   One can refer to https://github.com/google-research/google-research/tree/master/mutual_information_representation_learning and https://github.com/Philip-Bachman/amdim-public for the authors' official implementations, or checking the details in our released code.

**Consistent Trend on SimCLR [6]**   We also evaluate CPC, PCC, and D-RFC in SimCLR [6], which is a SOTA model and method on self-supervised representation learning. Note that the default contrastive learning objective considered in SimCLR [6] is CPC, which obtains $91.04\%$ test accuracy on CIFAR-10 (average for 5 runs). Details can be found in https://github.com/google-research/simclr. Similar to our **shallow** and **deep** experiments, we only change the contrastive learning objectives in SimCLR, and observing $91.51\%$ and $88.69\%$ average test accuracy for D-RFC and PCC, respectively. The trend is consistent with our **deep** experiment, where D-RFC works slightly better than CPC and PCC works slightly worse than CPC.

Figure 1: **Dataset Debugging** task with unsupervised word features across acoustic and textual modalities. *Probabilistic Classifier* approach is used to estimate PD between the audio and textual feature of a given word. The estimator is trained on the training split. We plot the logarithm of PD (i.e., PMI) distribution for the training words. We select the words with negative PMI values and categorize them into two groups: one contains the words end in "ly" and another containts the words end in "s".

## 4    More on Cross-Modal Learning

**Another Case Study: Cross-modal Adversarial Samples Debugging**    One important topic in interpretable machine learning [17] is dataset debugging, which detects adversarial samples in a given dataset. For instance, in this dataset, an adversarial word feature would have low statistical dependency between its audio and textual representations. In Fig. 1, we report the PMI distribution and highlight the training words with PMI $< 0$ (i.e., the adversarial samples). We note that a negative PMI means the audio and textual features are either statistically independent or even co-occur less frequently than the independent assumption.

First, we find the distribution of PMI resembles a Gaussian distribution. The mean of the PMI values is MI, and our empirical estimation for it is 8.37. Our goal is to identify the training samples with PMI that deviates far from MI, and especially for the samples with negative PMI. There are 147 words have negative PMI values, approximately 0.45% of the training words. Next, we select some of these words and categorize them into two groups. The first group contains the words end in "ly" and another group contains the words end in "s". That is to say, the words end in "ly" and "s" are adversarial training sample in our analysis. To sum up, we demonstrate how our PD estimation approach can be used to detect adversarial training examples in a cross-modal dataset.

**Dataset**    We construct a dataset that contains features from Word2Vec [16] and Speech2Vec [7]. Word2Vec is an unsupervised word embedding learning technique that takes a large text corpus of text as input and produces a fixed-length vector space. Specifically, each word in the corpus is assigned a real-valued and fixed-dimensional feature embedding. Similar to Word2Vec, Speech2Vec takes a large corpus of human speech as input and produces a fixed-length vector space. Specifically, it transforms a variable-length speech segment (a word in the speech corpus) as a real-valued and fixed-dimensional feature embedding. There are $37,622$ words shared across Word2Vec and Speech2Vec, where we consider $32,622$ words of them (randomly selected) to be the training split and $5,000$ of them to be the test split. That is to say, each word contains a textual feature (from Word2Vec) and an audio feature (from Speech2Vec), with both feature being $100-$dimensional. The dataset can be downloaded from <https://github.com/iamyuanchung/speech2vec-pretrained-vectors> and we include the training/test split in our released code.

**Training and Architectures**  We adopt the "separate critic" design [20, 22, 23] for our neural network parametrized function. Suppose $\hat{l}_\theta$ is the logits model in Probabilistic Classifier approach, and the separate critic design admits $\hat{l}_\theta(x, y) = g_{x\theta}(x)^\top g_{y\theta}(y)$ with $g_{x\theta}$ and $g_{y\theta}$ being different multiple layer perceptrons. We consider $g_{x\theta}$ and $g_{y\theta}$ to be 1-hidden-layer neural network with 512 neurons for intermediate layers, 128 neurons for the output layer, and ReLU function as the activation. The optimization considers batch size 512 and Adam optimizer [12] with learning rate 0.001. A sigmoid function is applied to $\hat{l}_\theta$ ($\hat{p}_\theta = \mathrm{sigmoid}(\hat{l}_\theta)$) to ensure $\hat{p}_\theta$ is a probabilistic output. We consider 100 training epochs.

**Reproducibility**  Please refer to our released code, where we also include the dataset and its training/ test split.

## 5  Practical Deployment for Expectation(s)

In practice, the expectations in Propositions 1, 2, 3, 4, 5, 6, and 7 are estimated using empirical samples from $P_{X,Y}$ and $P_X P_Y$. With mild assumptions on the compactness of $\Theta$ and the boundness of our measurement, the estimation error would be small by uniform law of large numbers [26].

## Footnotes

*Work done at Carnegie Mellon University.