[Reviews · NeurIPS 2020]

Review 1

Summary and Contributions: The paper focuses on developing methods for estimating point-wise dependency.

Strengths: The paper is well written, introducing all the required details and motivating the need to study point-wise dependency. The strength of the paper is in the experiments. 1. There are multiple different application scenarios studied, that are broad and representative. 2. The comparisons are extensive though they can be represented more convincingly (see below). 3. Self-supervised representation learning is a good useful application.

Weaknesses: The connection between Section 3.1 and Section 3.2 is hard to follow and can be written better. The paper discusses how PD can be naturally obtained when optimizing fro MI neural variational bounds but the part on these methods having large variance and hence the need for other methods for PD estimation could be motivated better. The proposed methods address an interesting problem but they follow from the density ratio method for PMI. More details on the novelty of the proposed approach can be helpful to the reviewer and some details in Section 3.1 can be abstracted and absorbed into related work as this is not really the paper's contribution. The results also can be represented and explained better. Especially connecting the high variance of the existing approaches and how the proposed approaches are better. For example, It is hard to understand how the results are better than SMILE in Figure 1. Given that the experiments are the major strength of the paper, this can be expressed more convincingly.

Correctness: The claims and methodology is interesting and correct.

Clarity: The paper is well-written and easy to read.

Relation to Prior Work: The paper clearly articulates its position with respect to the prior literature.

Reproducibility: Yes

Additional Feedback:


Review 2

Summary and Contributions: This paper focuses on estimating the point-wise dependency (PD) that measures the instance-level dependency between events taken by two random variables. The authors show that although PD can be obtained via optimizing mutual information (MI) neural variational bounds, it leads to large variance. The authors further propose two data-driven approaches to estimate PD: (1) Probabilistic Classifier and (2) Density-Ratio Fitting. The first one casts the problem into a binary classification by sampling data pairs from joint density as positive labels and from the product of marginals as negative labels. The second approach directly minimizes the expected square distance between the true and estimated PD. The authors applied their PD estimation method in several applications, including MI estimation (by plugging-in the point-wise MI obtained by taking the log of PD), self-supervised representation learning (by using the constructive learning approach, i.e., similar pairs having higher PD) and cross-model retrieval (using audio and text data). The proposed method was shown empirically comparable to the baselines.

Strengths: Point-wise dependency estimation is an interesting yet understudied research issue. I am glad to see efforts beyond just estimating the aggregated MI. The problem studied and the approaches taken seem pretty novel to me and are technically sound. The paper is well-written and organized. Intuitions are given as well as rigorous mathematical descriptions, which makes it very easy to follow, and I find it enjoyable to read. Besides, the evaluations, theoretical analysis and relevant discussion are also done with high standards.

Weaknesses: 1. In Fig.1, it seems that the probabilistic classifier approach is better than the density-ratio fitting approach, as it has both smaller bias and variance. However, in Fig. 2 for another task, the density-ratio fitting is consistently better than all other approaches. I am wondering if authors have any insights to the differences between their performance in different tasks. 2. In the cross-modal learning section, no baselines were compared, and the density-ratio fitting was neither compared. While I understand that the main purpose is to showcase the usage of PD, cross-modality learning is potentially an important application of PD estimation, so I would suggest authors to compare against some SOTA baselines in this topic.

Correctness: The approaches developed are technically sound. Both theoretical analysis and empirical evaluations are present and solid.

Clarity: The paper is well-written, organized and extremely easy to follow.

Relation to Prior Work: Relevant prior works are properly cited, discussed and compared.

Reproducibility: Yes

Additional Feedback: Please kindly see the Weaknesses section.


Review 3

Summary and Contributions: This paper studies estimating point-wise dependency of data instance. For this purpose, two methods are proposed. Experiments on three tasks demonstrate the effectiveness of the proposed methods.

Strengths: 1. The problem is important to the NeurIPS community. 2. The method is theoretically sound. 3. The empirical evaluation is extensive.

Weaknesses: 1. The contribution is not significant, given existing neural method for density ratio estimation and point-wise mutual information estimation. 2. The experiment is mainly conducted on toy data.

Correctness: They are correct.

Clarity: The paper is well written.

Relation to Prior Work: The discussion is clear.

Reproducibility: Yes

Additional Feedback: This paper studies estimating point-wise dependency of data instance. For this purpose, two methods are proposed. Experiments on three tasks demonstrate the effectiveness of the proposed methods. The paper also has several weaknesses: 1. The contribution is not so significant. 1) This paper focuses on estimating point-wise dependency of data instance, and the proposed methods are principled and theoretically sound. Despite the merit, similar problems have been extensively studied in the machine learning literature. For example, many prior works estimate the density ratio of two distributions by using the conjugate form of f-divergence, and these methods can be used to estimate the point-wise dependency. Also, some other works try to use neural method to estimate point-wise mutual information, which are also able to estimate the point-wise dependency. Given these existing studies, the idea of the paper seems quite straightforward. 2) For point-wise dependency estimation, two methods are proposed. The first Probabilistic Classifier method optimizes a classifier, which is then used to estimate the point-wise dependency. However, I feel like this method is a direct extension of GAN and f-divergence for density ratio estimation. For the second Density-Ratio Fitting method, it is also inspired by a prior work. In this sense, this paper does not propose much new insight on methodology. 3) The paper also points out that the problem of point-wise dependency estimation can be solved by existing neural estimator of mutual information. From my understanding, the proposed methods share very similar methodology to these methods. I wonder what is the advantage of the proposed methods for estimating point-wise dependency over these related methods? 2. The experiment is only conducted on toy dataset. Although the experiment in the paper is extensive, where three tasks are considered, the experiment is only conducted on toy datasets. For example, in application 1, different methods are evaluated with correlated Gaussian distributions; in application 2, two small datasets MNIST and CIFAR are used. For application 1, it is possible to evaluate with some other distributions? For application 2, is it possible to evaluate on ImageNet? For application 3, is there any baseline method to compare against? ------------------------- Thanks the authors for the clarity on the contribution and the additional experimental results! Overall, this is a solid work and I lean towards an accept.


Review 4

Summary and Contributions: In this paper, the authors study how to efficiently and effectively perform point-wise dependency estimation by neural methods. The main contribution could be summarized as follows. C1. An interesting angle to address mutual information estimation is discussed. C2. Probabilistic classifier and density-ratio fitting are proposed to enable effective point-wise dependency estimation. C3. The value of point-wise dependency estimation is highlighted from empirical study.

Strengths: S1. The authors suggest interesting perspectives to approach point-wise dependency estimation. S2. Theoretical evidences are provided to enrich the discussion on mutual information estimation. S3. The author explore multiple applications to demonstrate the value in point-wise dependency estimation.

Weaknesses: W1. It is difficult to clearly see the concrete difference/impact brought by either probabilistic classifier or density-ratio fitting. It could be the presentation in Figure 1. Instead of being qualitative, the authors may make this comparison more quantitative. W2. The value of point-wise dependency estimation in cross-modal learning is a bit weak. The task discussed in Section 6 seems to be a typical classification or ranking problem. To this end, to better show the value of point-wise estimation, it is important to make comparison with state-of-the-art baselines. Otherwise, only feasibility could be claimed, leaving limited impact.

Correctness: In general, yes, but the empirical results presentation may need improvement.

Clarity: Yes

Relation to Prior Work: Yes

Reproducibility: Yes

Additional Feedback: In terms of self-supervised representation learning discussed in Section 5, how do similar pairs are decided in MNIST and CIFAR10?

[Author Response · NeurIPS 2020]

We thank reviewers for their valuable feedback.

**General Response:** ▶ **(R1/R3) Contributions and Difference from Related Work.** Our first contribution is presenting PD estimation methods that avoid optimizing MI (neural) variational bounds. The probabilistic-classification method defines a binary-cross-entropy loss to differentiate samples from joint distribution or samples from the product of marginal distributions. GAN instead distinguishes between samples from true data distribution or samples from generator distribution. The density-ratio fitting method aims at estimating density-ratio between the joint distribution and the product of marginal distributions. Prior work studied kernel-based methods, while we take advantage of high-capacity neural networks.

Our second contribution is leveraging PD into 1) MI estimation, 2) self-supervised learning, and 3) cross-modal learning. For MI estimation, we find plugging-in estimated PD results in lower bias/variance than optimizing from MI's lower bound. We find the loss inspired by the density-ratio fitting method consistently outperforms the SOTA baseline for self-supervised learning. For cross-modal learning, we showcase the usage of PD for cross-modal retrieval (and cross-modal adversarial samples debugging provided in Supplementary) task(s).

▶ **(R1/R3) Advantages over MI Variational Bounds Methods.** We discussed the reason why we prefer the presented approach over the MI variational bounds methods in lines 125-131. It might seem that our idea is straightforward in retrospect, but we argue that its effectiveness in reducing the variance in MI variational bounds is important in many real-world applications. For example, prior work *[Song et al., Understanding the Limitations of Variational Mutual Information Estimators, 2019]* has pointed out that these MI variational bounds methods often have a large variance, and the large variance leads to the numerical issues in practice. On the contrary, our presented method does not estimate mutual information directly. We estimate the point-wise mutual information. Our approach either 1) utilizes the binary cross-entropy loss that has the benefit of numerical stability form the recent optimization package (e.g., PyTorch or TensorFlow) or 2) contains no logarithm or exponentiation, which is the cause of the numerical instability. In the final version of the paper, we will include more motivations for our presented approaches in Section 3.2.

▶ **(R1/R4) MI Estimation.** The comparisons with the SMILE method can be better understood by providing quantitative comparisons. For the quantitative analysis based on the bias-variance trade-off (e.g., $\text{Bias}^2$ + Variance), the Probalistic Classifier has the best performance, and the SMILE method is runner-up. The detailed quantitative numbers will be provided in the final version of the paper.

We would like to emphasize the that main takeaway from Figure 1 is an overall trend: estimating mutual information directly from its lower bound has a larger variance (with SMILE as an exception) compared to approximating mutual information by plugging-in the estimated PD. SMILE achieves superior performance because it clips the model's outputs to prevent abrupt large or small numbers. The remaining approaches in Figure 1 do not post-process the model's outputs. We will also include this discussion in the final version of the paper.

▶ **(R2/R4) Remark on Crossmodal Retrieval.** For this experiment, our purpose is to showcase the usage of PD estimation. Note that we have performed analysis using Density-Ratio Fitting method (92.26% top-1 retrieval accuracy) in line 280. In the future work, we will elaborate more on the usage of PD estimation for cross-modal retrieval and compare it with more baselines. Besides this experiment, in Supplementary, we also study cross-modal adversarial samples debugging using PD estimation.

**Reviewer #1:** ▶ **Connection between Section 3.1 and 3.2.** We thank R1 for suggesting 1) better motivating why we need the proposed method in Section 3.2 and 2) absorbing and abstracting some part of Section 3.1, which is not the main contribution of the paper. To address the concerns, we will slightly shorten Section 3.1 and expand Section 3.2 in the final version of the paper.

**Reviewer #2:** ▶ **Figure 1 and 2.** Figure 1 presents the results for MI estimation. Figure 2 shows the results for the self-supervised representation learning. Prior work *[Tschannen et al., On Mutual Information Maximization for Representation Learning, 2019.]* suggests a higher MI does not result in a better representation, which shares a similar observation as ours. A better understanding of the relationship between good MI estimation and a good representation learning objective is still an open research problem.

**Reviewer #3:** ▶ **Datasets are Toy Data.** For experiment 1, correlated Gaussians have a closed-form mutual information expression, and hence it is viewed as the benchmark experiment for mutual information estimation *[Belghazi et al., MINE: Mutual Information Neural Estimation, 2018]*. For experiment 2, we are happy to provide experiment on ImageNet. We use ResNet-50 as the backbone model, where we obtain the test accuracy 74.0% when using Density-Ratio Fitting method. The baseline is contrastive predictive coding, which gives us test accuracy 73.70%. We will include this result in the final version of the paper. For experiment 3, our purpose is to showcase the usage of PD estimation. It is our future plan to elaborate more on the usage of PD estimation for cross-modal retrieval and compare it with more baselines.

**Reviewer #4:** ▶ **Similar Pairs in MNIST/ CIFAR10.** For an input image, we perform two different data augmentations on this image, viewing these two augmented variants as a similar pair.

[Meta-Review · NeurIPS 2020]

The paper present some clear novelty in the under-studied domain of point-wise dependency estimation, with clear supporting evidence though various experiments. The reviewer concerns were clarified in the rebuttal, but we expect the authors to work significantly by including the changes they mention in their rebuttal, e.g: - shorten Section 3.1 and include more motivations for our presented approaches in Section 3.2. - include the MI discussion, - include ImageNet results in Experiment 2